# Model building of protein complexes from intermediate-resolution cryo-EM maps with deep learning-guided automatic assembly

Jiahua He [1], Peicong Lin[1], Ji Chen[1], Hong Cao[1] & Sheng-You Huang [1✉]

Advances in microscopy instruments and image processing algorithms have led to an increasing number of cryo-electron microscopy (cryo-EM) maps. However, building accurate models into intermediate-resolution EM maps remains challenging and labor-intensive. Here, we propose an automatic model building method of multi-chain protein complexes from intermediate-resolution cryo-EM maps, named EMBuild, by integrating AlphaFold structure prediction, FFT-based global fitting, domain-based semi-flexible refinement, and graph-based iterative assembling on the main-chain probability map predicted by a deep convolutional network. EMBuild is extensively evaluated on diverse test sets of 47 single-particle EM maps at 4.0–8.0 Å resolution and 16 subtomogram averaging maps of cryo-ET data at 3.7–9.3 Å resolution, and compared with state-of-the-art approaches. We demonstrate that EMBuild is able to build high-quality complex structures that are comparably accurate to the manually built PDB structures from the cryo-EM maps. These results demonstrate the accuracy and reliability of EMBuild in automatic model building.

[1] School of Physics and Key Laboratory of Molecular Biophysics of MOE, Huazhong University of Science and Technology, Wuhan, Hubei 430074, China. ✉email: huangsy@hust.edu.cn

Advances in cryo-electron microscopy (cryo-EM) instruments, data collection, and image reconstruction[1–8] have led to an increasing number of cryo-EM density maps of previously intractable biological systems[9,10]. However, the goal of cryo-EM is not to reconstruct density maps but to determine the atomic structures. For high-resolution maps (<3 Å), high-quality atomic structures can be built using the software conventionally designed for X-ray crystallography[11,12]. For cryo-EM maps with resolutions <4 Å, de novo model building[13–16] also achieves a satisfactory performance. However, for cryo-EM maps at intermediate resolutions (4–8 Å), building accurate structure models remains a challenging and labor-intensive process, which is reflected in the giant gap between the numbers of reconstructed density maps at intermediate resolutions and the deposited three-dimensional (3D) structures in the protein data bank (PDB). As of 1 January 2022, a total of 3746 cryo-EM maps with resolutions ranging from 4.0 to 7.9 Å are deposited in EMDB[17], but only 2218 of which have associated PDB[18] structures, which means that there is no available structure for >40% of intermediate-resolution cryo-EM maps. Most of these uninterpreted maps are solved by single-particle cryo-EM. However, with the rapid development of cryo-electron tomography (cryo-ET), intermediate resolution maps obtained by subtomogram averaging of cryo-ET data become more widely available[19–23]. As such, methods for accurate structural interpretation of intermediate resolution EM maps are urgently in demand.

Although some efforts have been made to make up for the gap[24–27], building a model from scratch is challenged by massive uncertainty for the cryo-EM maps at intermediate resolutions. Prior knowledge is normally required for such types of model building, which in most cases starts from given initial template structures. Then, the atomic model of an EM map can be built by fitting and refining the template structure against the map. The initial template structures may be taken from previously solved high-resolution structures, or predicted through structure prediction methods like homology modeling, fragment threading, and deep learning. Rigid fitting and flexible fitting are common techniques to place a template structure into intermediate resolution cryo-EM maps. Rigid fitting searches for possible relative orientations between a structure and a density map. The fitness between the fitted structure and the map is measured by a scoring function, e.g., cross-correlation, mutual information, SCCC, SMOC[28], etc. Until now, various rigid fitting tools have been developed, including EMfit[29], UCSF Chimera[30], gmfit[31,32], multifit[33,34], Situs[35], PowerFit[36], TEMPy[37–39], MOFIT[40], VESPER[41], and Phenix[42]. If the starting template structure exhibits a certain degree of deviation from the ground truth structure, flexible fitting[43–50] is often required to improve the rigidly fitted structure to conform to the density map.

Although many significant milestones have been reached, existing algorithms still have limited accuracy in structure determination from intermediate-resolution cryo-EM maps due to several challenges. First, experimentally solved EM maps usually contain heterogeneous density signals and random noises. Therefore, scoring functions that measure the fitness between the structure and experimental map may mislead the searching and ranking procedures of the fitting. Current methods often have no alternative but to pursue robustness at the sacrifice of scoring accuracy. Second, human intervention is still necessary for the majority of fitting algorithms, which makes fitting labor-intensive and extremely unfriendly for non-expert users. The fitting results are severely affected by specific parameters. Third, many methods are designed for single-chain protein fitting and therefore require map segmentation of individual subunits. Complex structures can only be built by manually combining multiple fitting results of individual chains. In such case, accurate map segmentation of individual subunits is impossible because of the low quality of the

map, let alone reliable fitting of individual chains into the map. Finally, proteins are flexible molecules and thus flexible fitting is often required during model building, which is a painstaking procedure. For molecular dynamics-based refinement approaches, the initialization is complicated, the calculation is time-consuming, the result is parameter-dependent, and the entire procedure may suffer from errors. As such, lightweight refinement protocols are more commonly used, but they cannot deal with large conformational changes.

Addressing the challenges, we develop a deep learning-guided method to automatically build the structure of multi-chain protein complexes from intermediate-resolution cryo-EM maps, which is referred to as EMBuild. Through iteratively fitting, refining, and assembling of protein structures of individual chains predicted from sequences, EMBuild can build high-quality protein complex structures without human intervention. Instead of directly fitting protein chains to the original density map, EMBuild fits the chains to the main-chain probability map predicted by our deep learning model, where the density value on a grid point stands for the probability of finding a main-chain atom around the grid point. Compared with the density map, the main-chain probability map includes more precise location information of main-chain atoms, which can much help improve the accuracy of fitting. Following rigid fitting, a semi-flexible domain refinement strategy is implemented, which performs fast optimization of domain orientations. The final protein complex structure is assembled using a graph-based search of the top-scored combination of fitted protein chains. We evaluate the performance of EMBuild on diverse test sets of 47 single-particle EM maps at 4.0–8.0 Å resolution and 16 subtomogram averaging maps at 3.7–9.3 Å resolution. Our results show that the model of the protein complex structure, i.e., complex model, built by EMBuild is of high quality with respect to not only the reference PDB structure but also the EM density map in terms of various metrics. In addition, we also demonstrate that EMBuild can also reliably estimate the quality of the built models.

## Results

**Overview of EMBuild.** For EMBuild, our goal is to automatically build the protein complex structure from a given EM density map starting from sequences. Figure 1 shows an overview of the EMBuild workflow. The input of EMBuild is a density map and the corresponding protein sequences of individual chains. From the input density map, the main-chain probability map is predicted by EMBuild using a nested U-net (UNet++)[51] that was trained on a set of 209 pairs of experimental density maps and main-chain probability maps calculated from deposited PDB structures (Supplementary Data 1). Given the input protein sequences of individual chains, their 3D structures are modeled by a protein structure prediction program. Here, AlphaFold2[52] (AF2) is used to predict the protein structures from sequences, though other programs like I-TASSER and Rosetta could also be used. Then, each predicted protein structure is fitted to the main-chain probability map using a fast Fourier transformation (FFT)-based global alignment method. To consider a certain degree of deviation between the input protein chain model and ground truth structure, we adopt a semi-flexible domain refinement strategy in EMBuild. Namely, the input protein chain model is first optimally fitted as a rigid entirety. Then, each structure domain of the fitted protein chain is locally refined, as illustrated in Fig. 2a. For each fitted protein chain, we use a scoring function to measure how well it matches the main-chain probability map, referred to as the main-chain match score. With all the fitting results of individual protein chains, the final protein complex structure is selected from different combinations of fitted

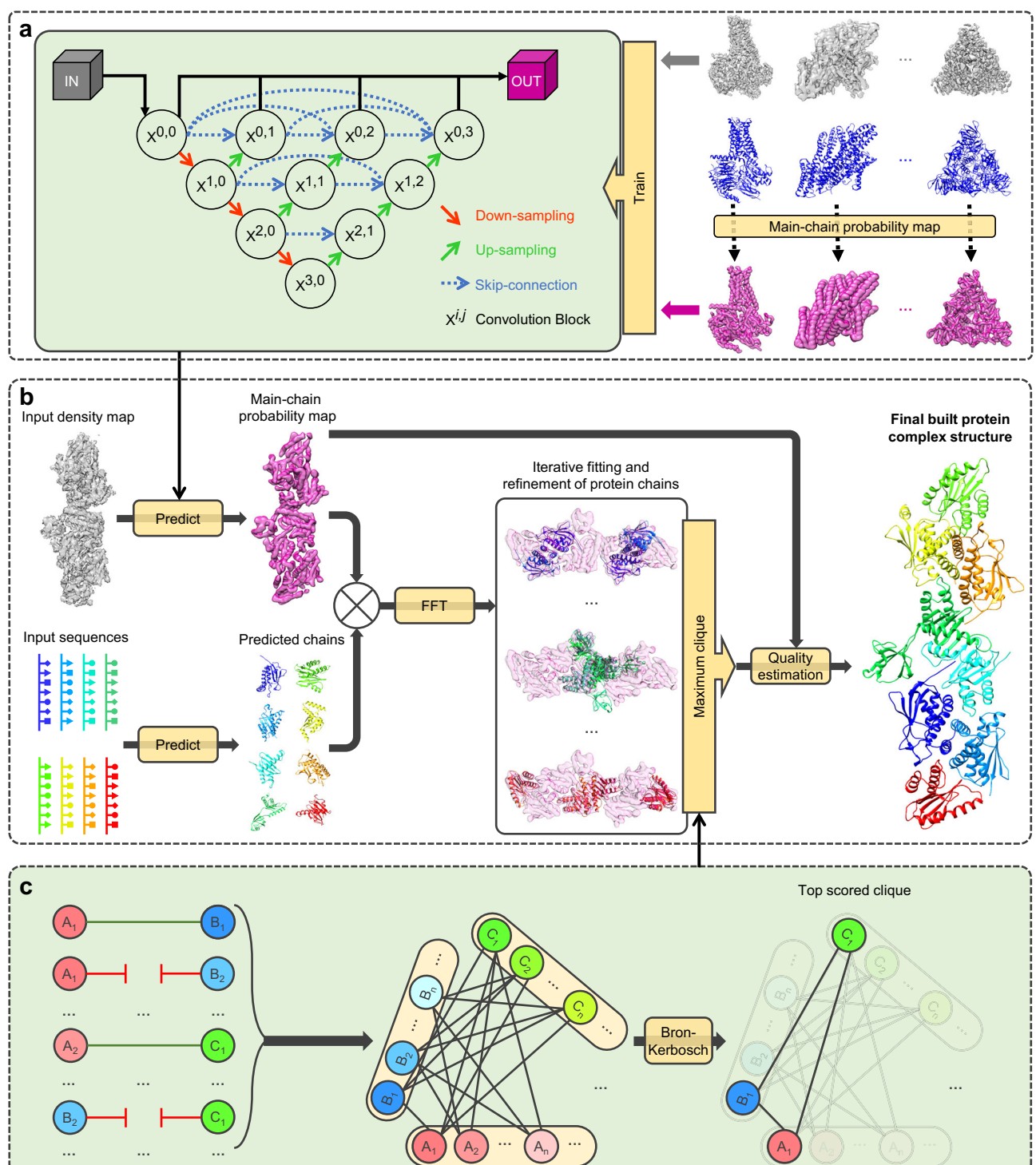

**Fig. 1 Overview of the EMBuild workflow. a** The training strategy of the deep learning module for main-chain probability prediction. The deep learning module adopts a UNet++[51] architecture. The training set consists of pairs of experimental EM maps and main-chain probability maps computed from associated PDB structures. **b** The workflow of EMBuild. The input of EMBuild is the EM density map and its corresponding sequences of individual protein chains. The main-chain probability map is predicted from the input density map using our trained deep learning model. The atomic models of individual chains are predicted from their sequences by a structure prediction program. A set of predicted chain models are individually fitted to the main-chain probability map through an FFT-based fitting and refinement. With all the fitting results of individual chains, the final protein complex structure is built through a Bron-Kerbosch maximum clique algorithm. **c** Schematic of the Bron–Kerbosch maximum clique algorithm. The clash scores between two fitted poses from different chains are calculated. Two poses are connected (green line) if the clash score is below a certain threshold, and are disconnected (red break line) if the clash score is above the threshold. Then, a graph is generated where each vertex has the fitting score of its associated pose. The Bron-Kerbosch search algorithm is used to find the top-scored combination of chains that are adjacent to each other, i.e., the top-scored clique.

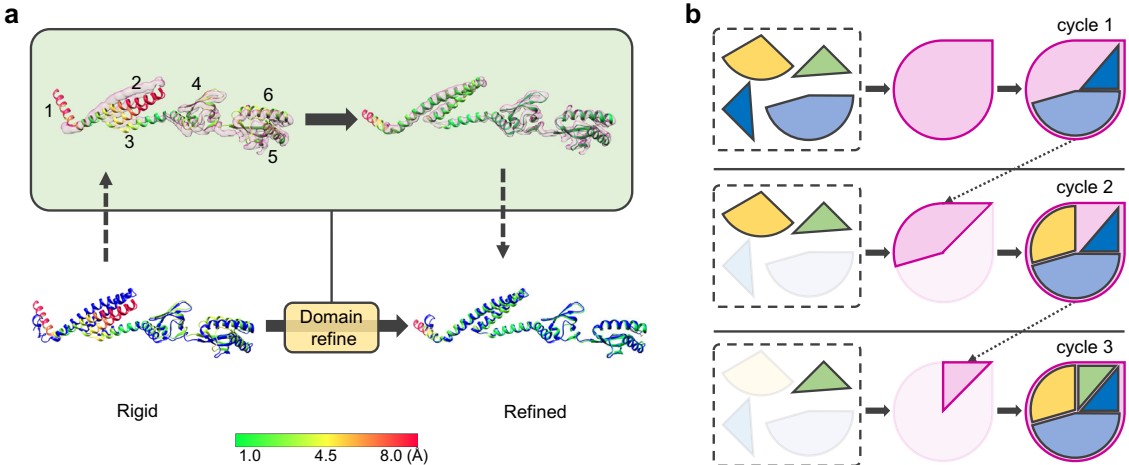

**Fig. 2 Detailed illustration of iterative flexible fitting. a** Schematic of the domain refinement procedure applied in EMBuild. The main-chain probability map is in transparent magenta, the reference PDB structure is in blue, and the fitted query model is colored from green to red according to the Cα displacements between the query model and the PDB structure. Owing to the conformational deviation between the query model and the PDB structure, the query chain matches poorly to the reference structure as a rigid entirety (bottom left). For better fitting, the query chain is then split into multiple domains, as indicated by numbers 1–6 (top left). At last, all domains are iteratively optimized to achieve their best fitting into the main-chain probability map (top right). As such, the final fitted model shows good consistency with the PDB structure (bottom right). **b** The iterative building strategy adopted by EMBuild. After each cycle of building, the map regions with fitted structures are removed from further fitting. The remaining chains are iteratively assembled to the complex under the guidance of the updated main-chain probability map.

positions. Specifically, EMBuild applies a Bron–Kerbosch maximum clique algorithm to select the best combination of protein chains with the highest total main-chain match score among different combinations, in which severe atomic clash between chains is not allowed. However, there will be a possibility that some chains cannot be assembled to the complex through only a single cycle of Bron–Kerbosch algorithm. Therefore, EMBuild adopts an iterative assembling strategy to improve the integrality of built protein complexes, as indicated in Fig. 2b. Finally, the quality of the built model is estimated using the main-chain match score calculated between the modeled structure and the main-chain probability map.

**Evaluating built models against the PDB structure**. EMBuild was first evaluated on the test set of 47 experimentally solved single-particle cryo-EM maps of protein complexes, and compared with phenix.dock_in_map[42], DEMO-EM[53], and gmfit[31,32]. For all the methods, the input structures of protein chains are predicted from sequences by AlphaFold2[52]. It should be noted that DEMO-EM was evaluated in two ways. One is using sequences as the input. The other is using AlphaFold2-predicted structures as the input. To evaluate the accuracy of the built models, the built models were aligned to the deposited PDB structures using MMalign[54], which gave the values of two metrics, TM-score and RMSD. Here, the TM-score is a measure of similarity between two protein structures[55], and the RMSD stands for the root mean square deviation of the aligned residues between the built complex model and the PDB structure. Figure 3 shows the TM scores and RMSDs of the built complex models with respect to the PDB structures by different approaches. The detailed results for each test case are listed in Supplementary Data 2.

It can be seen from Fig. 3a that EMBuild significantly outperformed the other methods. Specifically, the complex models built by EMBuild achieved an average TM-score of 0.909, which is significantly higher than 0.746 for phenix.dock_in_map, 0.532 for DEMO-EM, 0.631 for DEMO-EM with AlphaFold2 structures, and 0.515 for gmfit. EMBuild also achieved a better performance than the other methods on most of the test cases (Fig. 3b). Encouragingly,

EMBuild has succeeded in building accurate complex models with TM-score > 0.5 for all of the 47 test cases. In addition to having a better TM-score, the complex models built by EMBuild also achieved an average RMSD of as low as 2.85 Å, which is significantly better than 4.52 Å for phenix.dock_in_map, 5.11 Å for DEMO-EM, 4.37 Å for DEMO-EM with AlphaFold2 structures, and 6.64 Å for gmfit (Fig. 3c). Compared with the other methods, EMBuild can build better complex models on most of the test cases in terms of RMSDs (Fig. 3d). Especially, EMBuild has succeeded in building accurate complex models with RMSD < 6 Å for all of the 47 test cases.

From Fig. 3, we can also observe two other notable features. One feature is that DEMO-EM + AF2 performed better than DEMO-EM in both TM-score (0.631 vs. 0.532) and RMSD (4.37 Å vs. 5.11 Å). As the only difference between DEMO-EM + AF2 and DEMO-EM lies in the input initial models for assembling, the better performance of DEMO-EM + AF2 than DEMO-EM will be attributed to the more accurate models built by AlphaFold2 than by I-TASSER. The other feature is that EMBuild performed much better than DEMO-EM + AF2 in both TM-score (0.909 vs. 0.631) and RMSD (2.85 Å vs. 4.37 Å). As EMBuild and DEMO-EM + AF2 adopted the same AlphaFold2 models as input but different assembling strategies, the much better performance of EMBuild than DEMO-EM + AF2 would come from the more advanced assembling strategy in EMBuild than in DEMO-EM + AF2. These results suggest that the high accuracy of EMBuild is not only because of the better input models of protein chains but comes much more from the advanced assembling strategy in EMBuild.

Figure 4 shows several examples of the complex models built by EMBuild on single-particle EM maps. The first example is EMD-3605, which is a 4.2 Å cryo-EM map for the full-length structure of ZntB. Figure 4a shows a comparison between the EMBuild model and the PDB structure on EMD-3605. It can be seen from the figure that the EMBuild model reproduced the conformation in the deposited structure of the pentamer complex structure. The model built by EMBuild achieved the best quality among different methods. Specifically, the EMBuild model yielded a TM-score of 0.986 and an RMSD of 1.63 Å, compared

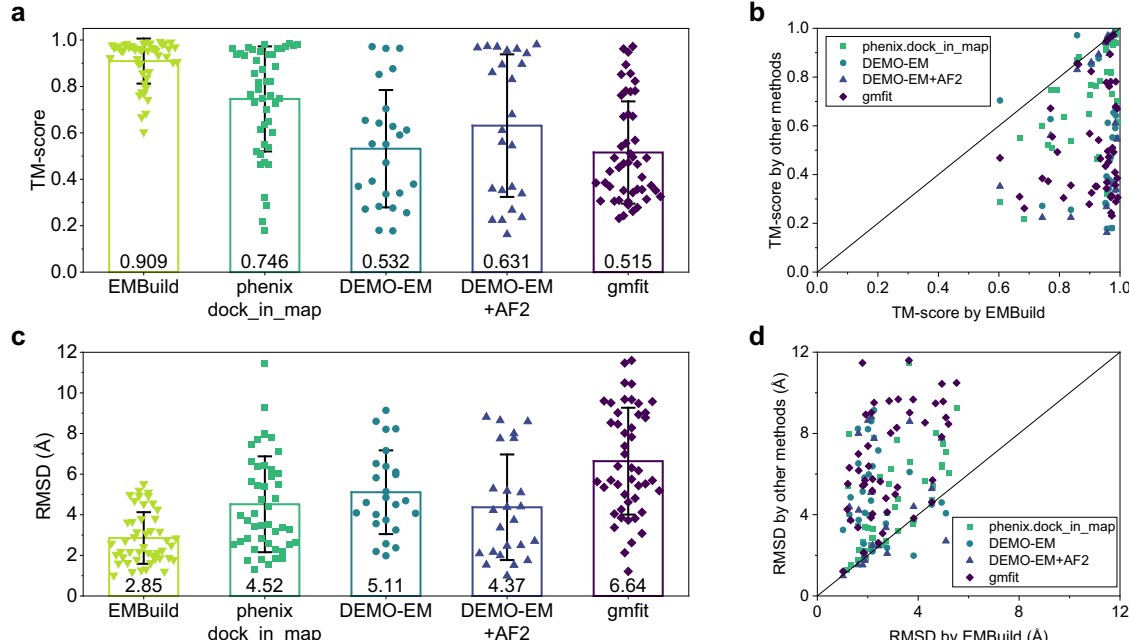

**Fig. 3 Evaluation of the built models against the reference PDB structure for EMBuild, phenix.dock_in_map, DEMO-EM, DEMO-EM with AlphaFold2 structures, and gmfit, on the test set of 47 single-particle EM maps. a**, **c** Average TM-scores (**a**) and RMSDs (**c**) of the built models for $n = 47$ individual test cases. Error bars indicate ±1.0 standard deviation. **b**, **d** Comparison of the TM-scores (**b**) and RMSDs (**d**) on each test case between EMBuild and other methods. Source data are provided in the Source Data file.

with 0.939 and 3.45 Å for phenix.dock_in_map, 0.642 and 6.52 Å for DEMO-EM, 0.338 and 8.00 Å for DEMO-EM with AlphaFold2 structures, and 0.289 and 6.99 Å for gmfit. In addition, the EMBuild model gave low Cα displacements in the entire model including the transmembrane domain and the cytoplasm domain. Nevertheless, there is still room for improvement on some flexible regions, such as the coils in the periplasm domain, as well as some linkers between helices and strands.

EMbuild can also be applied to virus proteins. As illustrated in Fig. 4b, EMBuild succeeded in building a high-quality model for EMD-3856, which is the (foot and mouth disease virus) FMDV A10 inside-out particle. The trimer model built by EMBuild achieved a TM-score of 0.960 and an RMSD of 1.80 Å. In the built model, EMBuild recovered almost all the residue positions in the PDB structure except for some loop regions (indicated by arrow). Another example of virus protein is EMD-6685 for the Japanese encephalitis virus. The EMBuild model for this map also showed a perfect agreement with the PDB structure, which can be seen from Fig. 4c. The hexamer model for EMD-6685 built by EMBuild achieved a TM-score of 0.985. This is a relatively easy target since the protein chain models predicted by AlphaFold2 are close to their PDB structures. As such, phenix.dock_in_map was also capable to build a high-quality structure with a TM-score of 0.973. Nevertheless, EMBuild achieved a bigger advantage in the RMSD of the build model, and gave an RMSD value of 1.64 Å, which is significantly better than 2.28 Å for phenix.dock_in_map.

For other kinds of protein complexes, EMBuild also showed consistently good performances. This is true for EMD-7453, which is a complicated hetero-octamer involved in tetherin downregulation, as illustrated in Fig. 4d. It should be noted that two short chains (Chain T with 13 residues and Chain L with 10 residues) are ignored in the modeling and evaluating processes of EMD-7453. EMD-7453 is an extremely hard target because there exist large conformational changes between the predicted chain structure by AlphaFold2 and the PDB structure. Nevertheless, the EMBuild model still recovered the conformation of the deposited structure with a TM-score of 0.992 and an RMSD of 1.21 Å, which drastically exceeded 0.616 and 5.39 Å for phenix.dock_-in_map, and 0.306 and 5.52 Å for gmfit, respectively. Another good model was constructed by EMBuild for EMD-9317 with eight chains, as shown in Fig. 4e. The EMBuild model achieved a TM-score of 0.983 and an RMSD of 1.92 Å. It can be revealed from the figure that the EMBuild model exhibited low Cα displacements in the entire model except for those regions with weaker density signals. In addition, EMBuild is also capable of building protein complexes with more chains, taking EMD-22216 for instance. As displayed in Fig. 4f, ten protein chains were correctly assembled by EMBuild, resulting in a high-quality complex model with a TM-score of 0.970 and an RMSD of 2.04 Å.

**Validating built models against cryo-EM maps**. We have evaluated the performance of EMBuild using the PDB structure as the reference. However, in real applications, the ground truth structure for a given map is commonly unknown. Therefore, as an alternative, it is of vital importance to evaluate how the built model represents the given map, i.e. fit-to-map of the built models[56]. Correspondingly, we reported the CC_box and CC_mask values calculated by phenix.map_model_cc, and map-model FSC05 values calculated by phenix.mtriage between the built model and the density map[57] (Supplementary Data 2).

Figure 5a shows the average CC_box of the built complex models by different methods. It can be seen from the figure that the complex models built by EMBuild achieved a significantly higher correlation coefficient with the EM map than the other methods. Specifically, EMBuild achieved an average CC_box value of 0.7152, compared with 0.5975 for phenix.dock_in_map, 0.5647 for DEMO-EM, 0.6077 for DEMO-EM with AlphaFold2 structures, and 0.5332 for gmfit. In addition, the average CC_box (0.7152) for the EMBuild models reaches near 95% of that (0.7545) for the PDB structures, which suggests that the EMBuild models can to some extent rival the quality of the

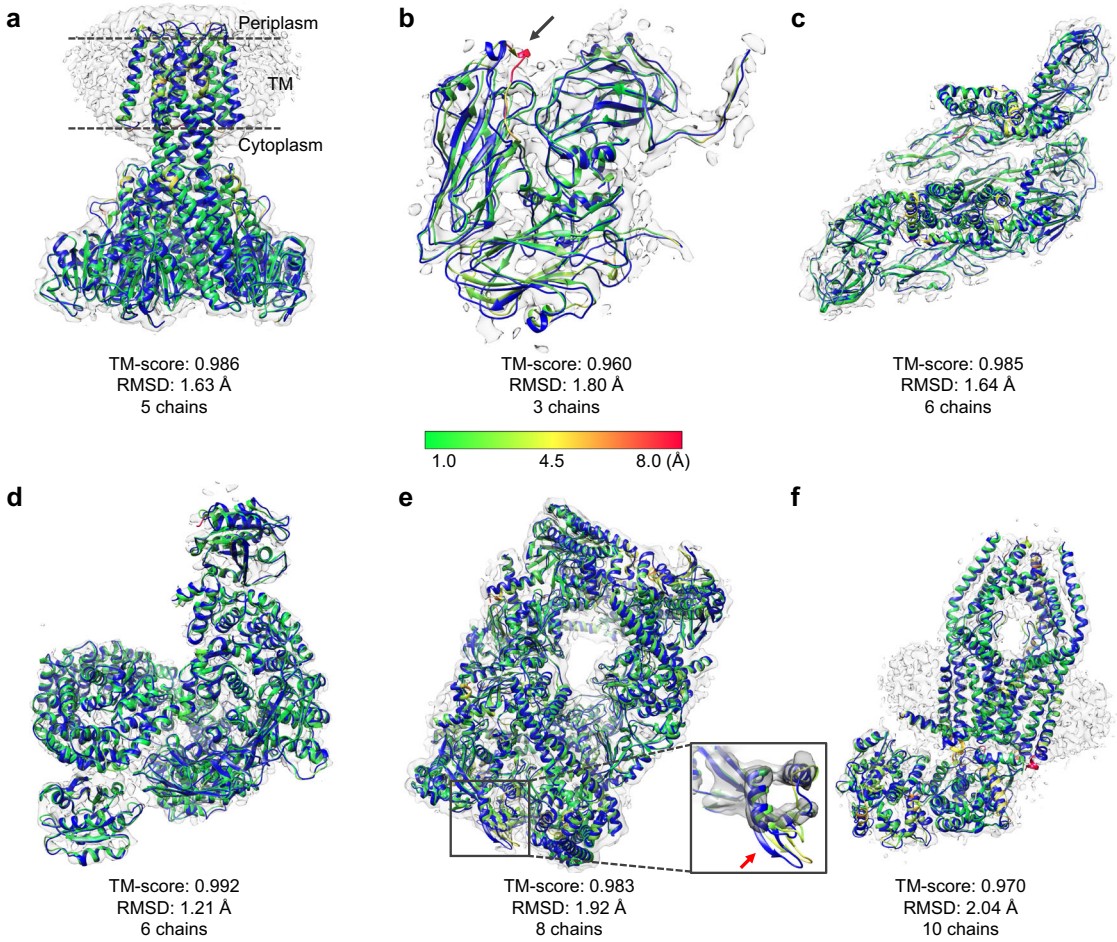

**Fig. 4 Examples of the protein complex structures built by EMBuild on single-particle cryo-EM maps.** The reference PDB structures are colored in blue and the corresponding EM density maps are colored in transparent gray. The built structures by EMBuild are colored from green to red according to Cα displacements with respect to the reference structure. **a** EMD-3605 at 4.2 Å resolution (PDB ID: 5N9Y). **b** EMD-3856 at 5.2 Å resolution (PDB ID: 5OWX). A poorly modeled loop is indicated by an arrow. **c** EMD-6685 at 4.3 Å resolution (PDB ID: 5WSN). **d** EMD-7453 at 4.3 Å resolution (PDB ID: 6D83). The enlarged view displays the EM density volume around a poorly modeled region (indicated by red arrow). **e** EMD-9317 at 5.2 Å resolution (PDB ID: 6N1Q). **f** EMD-22216 at 4.6 Å resolution (PDB ID: 6XJX).

PDB models. As shown in Fig. 5b, the majority of models built by EMBuild achieved a higher CC_box value than the other methods. EMBuild succeeded in building models with a CC_box of >0.5 for 46 out of the total of 47 cases. Similar trends can also be found for the CC_mask values, as can be seen in Fig. 5c. Specially, the complex models built by EMBuild achieved an average CC_mask value of 0.6898, compared with 0.5323 for phenix.dock_in_map, 0.5167 for DEMO-EM, 0.5573 for DEMO-EM with AlphaFold2 structures, and 0.3939 for gmfit. The average CC_mask (0.6898) for the EMBuild models is also as high as 95% of that (0.7295) for the PDB structures, which demonstrated the superiority of EMBuild. A detailed comparison further showed the leading performance of EMBuild, as it outperformed the other methods on most of the test cases (Fig. 5d). Out of the 47 cases, 44 cases built by EMBuild achieved a CC_mask of >0.5.

Similar improvement trends can be observed in the map-model FSC05 values of built complex models. Specifically, the EMBuild models obtained an average FSC05 value of 6.40 Å on the test set of 47 cryo-EM maps, compared with 5.71 Å for the deposited PDB structures. In contrast, the other methods either yielded a significantly higher FSC05 value or failed to give a valid map-model FSC on many more cases because their built models do not conform to the map. Out of 47 test cases, the deposited PDB

structures and the EMBuild models only show one failed case in the FSC05 calculation, compared with 17 failed cases for phenix.dock_in_map, 7 failed cases for DEMO-EM, 6 failed cases for DEMO-EM with AlphaFold2 structures, and 34 failed cases for gmfit (Supplementary Data 2). These results demonstrated the reliability of the built models by EMBuild.

**Evaluations on subtomogram averaging maps of cryo-ET data.** EMBuild was further evaluated on the test set of 16 EM maps obtained by subtomogram averaging of cryo-ET data, and compared with phenix.dock_in_map[42] and gmfit[31,32]. Figure 6a, b shows the TM-scores and RMSDs of the built complex models with respect to the PDB structures. The detailed evaluation results for each of the 16 test cases are listed in Supplementary Data 3. As illustrated in Fig. 6a, b, EMBuild has significantly outperformed the other methods on most of the test cases in terms of TM-score and RMSD values. On average, the complex models built by EMBuild achieved an average TM-score of 0.863 and an average RMSD of 2.74 Å, respectively, which are significantly better than 0.600 and 5.12 Å for phenix.dock_in_map and 0.386 and 7.96 Å for gmfit. Figure 6c, d shows the CC_box and CC_mask values of the complex models built by different methods. It can be seen from the figure that the complex models built by EMBuild also achieved a significantly higher average correlation coefficient than

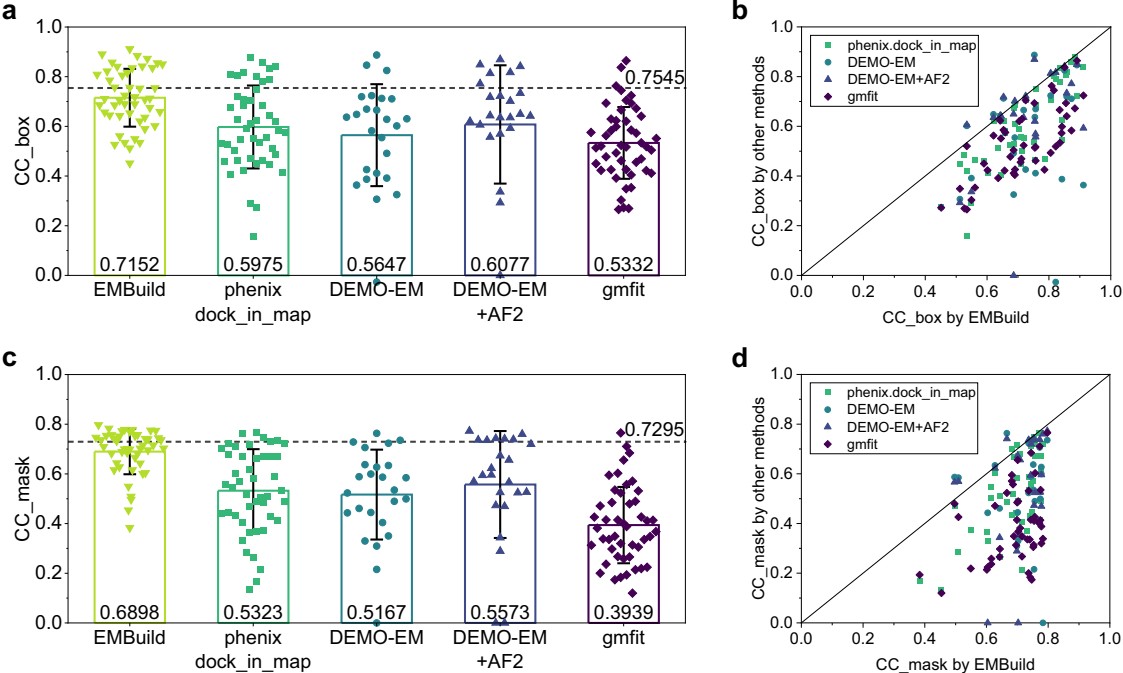

**Fig. 5 Evaluating map-model correlation coefficient (CC) of built models by EMBuild, phenix.dock_in_map, DEMO-EM, DEMO-EM with AlphaFold2 structures, and gmfit, on the test set of 47 single-particle EM maps. a, c** Average CC_box (**a**) and CC_mask (**c**) values among $n = 47$ individual test cases. Error bars indicate ±1.0 standard deviations. The dashed lines indicate the corresponding average CC values ($n = 47$ test cases) achieved by the PDB structures. **b, d** Comparison of the CC_box (**b**) and CC_mask (**d**) values on each test case between EMBuild and other methods, respectively. Source data are provided in the Source Data file.

the other methods on the test set of subtomogram averaging maps. Specifically, EMBuild achieved an average CC_box of 0.6887 and an average CC_mask of 0.6654, compared with 0.5639 and 0.5369 for phenix.dock_in_map and 0.4535 and 0.3638 for gmfit. Similar to the trends for the test set of single-particle EM maps, the average CC values of the EMBuild models are close to the average CC values of the deposited PDB structures (0.7628 and 0.6897 for CC_box and CC_mask, respectively).

Figure 6e, f shows two examples of the complex models built by EMBuild on the subtomogram averaging maps. The first example is EMD-10381, which is a 3.7 Å map for the EIAV CA-SP hexamer with 18 protein chains. The comparison between the EMBuild model and the PDB structure is displayed in Fig. 6e. It can be seen from the figure that EMBuild succeeded in building a high-quality model on this subtomogram averaging map. The model built by EMBuild achieved the best quality among different methods. Specifically, the EMBuild model yielded a TM-score of 0.996 and an RMSD of 1.09 Å, compared with 0.894 and 6.44 Å for phenix.dock_in_map, and 0.307 and 12.51 Å for gmfit. In terms of fit-to-map metrics, the EMBuild model achieved the CC_box, CC_mask, and map-model FSC05 of 0.6724, 0.7547, 3.75 Å, respectively, which are close to 0.7156, 0.7696, and 3.79 Å for the deposited PDB structure. The other example is EMD-3478 with 14 protein chains, as shown in Fig. 6f. Although the input subtomogram averaging EM map is at a poor resolution of only 8.0 Å, the EMBuild model still reproduced the conformation of the PDB structure with a TM-score of 0.991 and an RMSD of 1.19 Å, which drastically exceeded 0.175 and 3.38 Å for phenix.dock_in_map, and 0.274 and 9.72 Å for gmfit. In addition, the quality of EMBuild model is also comparable to that of the deposited PDB structure. The CC_box, CC_mask, and map-model FSC05 achieved by the EMBuild model are 0.8638, 0.7451, and 8.00 Å, respectively, compared to 0.8273, 0.6399, and 8.05 Å for the PDB structure.

**Quality check of built models.** Quality assessment is critical for model building into cryo-EM maps, so that users would know which parts are reliable and which parts need further check for the built model. Here, we propose the main-chain match score as a metric to indicate the quality of the built model, which measures the consistency between the main-chain atoms of the built model and the main-chain probability map (Eqs. (10) and (11)). We first investigated the relationship between the main-chain match scores and alignment scores of continuous secondary structure fragments with no less than 5 residues, which is shown in Fig. 7a. The alignment score is measured using the formula of Eq. (12) as a normalized distance between each pair of aligned modeled fragment and fragment in the PDB structure. It can be seen from the figure that the main-chain match score has a good correlation with the alignment score. The fragments with main-chain match scores of $<-12.0$ can normally achieve an average alignment score of above 0.7. Similar correlation and threshold can also be found at the residue level between the main-chain match scores and the quality of the built model. It should be noted that a poor main-chain match score does not necessarily mean a poor model. It could mean a large deviation from the true structure or a weak density in the region of the map, which just needs further examination.

Figure 7b shows an example of EMD-8794, which is the katanin hexamer in spiral conformation[58]. The EMBuild model achieved a TM-score of 0.979, an RMSD of 2.01 Å, a CC_box of 0.8577, and a CC_mask of 0.7569. Despite its high quality, some parts of the built model should be further checked, as indicated by their lower main-chain match scores. For example, the helices in the peripheral portion of the hexamer and the coils in the core show relatively lower match scores compared with other parts of the built model, which are consistent with the Cα displacements between the built model and the PDB structure. In addition, the regions around the opened interface of the first and last

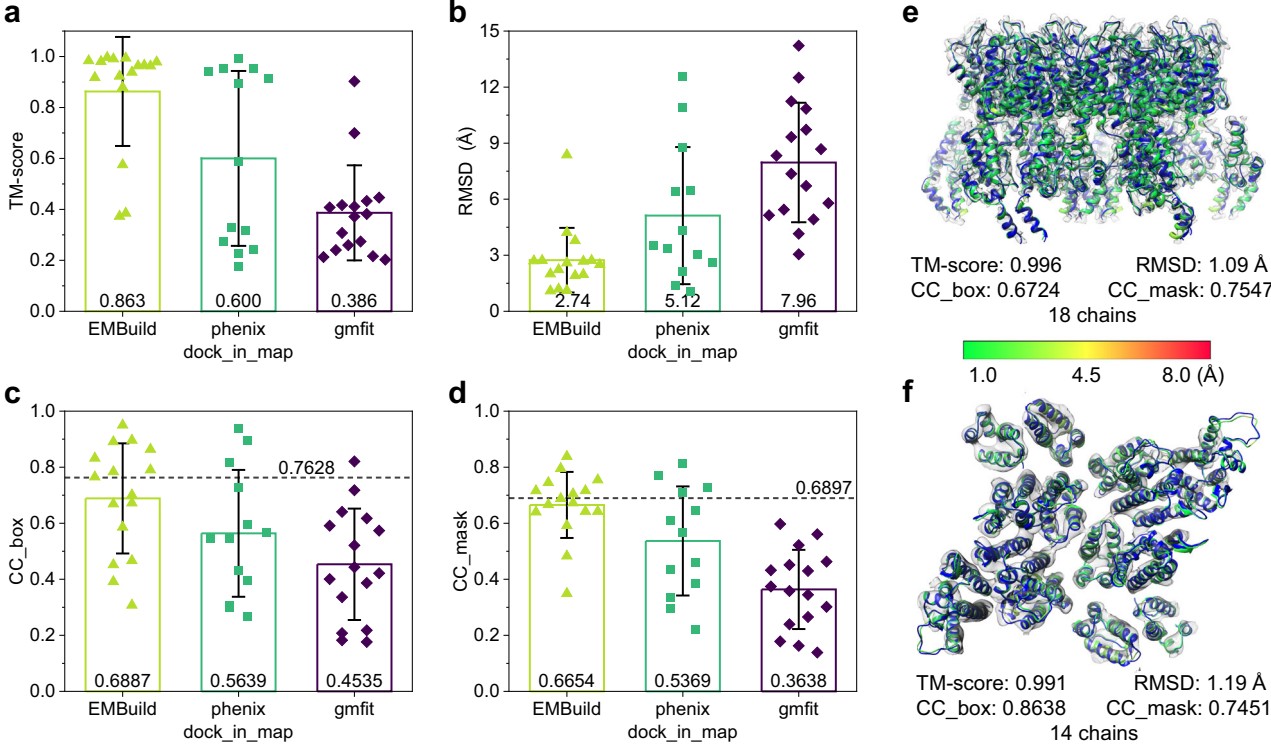

**Fig. 6 Evaluation results on the test set of 16 EM maps obtained by subtomogram averaging of cryo-ET data. a–d** Average TM-score (**a**), RMSD (**b**), CC_box (**c**), and CC_mask (**d**) values of the complex models built by EMBuild, phenix.dock_in_map, and gmfit ($n = 16$ individual test cases). Data are presented as mean values ± 1.0 standard deviation. The dashed lines indicate the corresponding average CC values achieved by the PDB structures ($n = 16$ test cases). **e, f** Examples of the protein complex structures built by EMBuild on the tomogram averaging maps for EMD-10381 at 3.7 Å resolution (**e**) and EMD-3478 at 8.0 Å resolution (**f**), respectively. The reference PDB structures are colored in blue and the corresponding EM density maps are colored in transparent gray. The built structures by EMBuild are colored from green to red according to Cα displacements with respect to the reference structure. Source data are provided in the Source Data file.

protomers (P1 and P6) also have lower match scores. Since P1 and P6 play the gating role in the cycling between the open spiral and closed ring conformations of katanin, the interface regions of P1 and P6 are dynamic and flexible, resulting in weak signals in density map, as shown in the enlarged view. On one hand, the helix bundle domain (HBD) of P1 (indicated by the top arrow) was not accurately placed by EMBuild due to lack of effective density and main-chain probability signal. On the other hand, in spite of the fact that a short loop of P6 (indicated by the bottom arrow) modeled by EMBuild has low Cα displacements with respect to the PDB structure, it should be further carefully validated in lack of clear density signal. These subtle details were precisely captured by the main-chain match scores.

Similar trends can also be observed in other examples displayed in Fig. 7c–e. For EMD-9631 in Fig. 7c, the poorly modeled coils can be located according to their low match scores. For EMD-20510 in Fig. 7d, detailed conformations for inter-subunit interactions (in the box) should be further improved. For EMD-20950 in Fig. 7e, the regions close to the C-terminal require further optimization, as indicated by the arrow. Encouragingly, the registration error of one inner helix can also be identified by the main-chain match score (Fig. 7e). In these examples, the main-chain match scores presented great consistency with the deviation between the modeled structure and the reference PDB model, and thus can be used as an indicator to guide further refinement and verification.

Furthermore, we assessed the quality of EMBuild-built models using coordinates-only metrics. Specifically, we examined the Ramachandran scores and MolProbity score calculated by MolProbity[59]. The detailed results are listed in Supplementary

Data 4. It can be seen from the table that EMBuild built high-quality models and achieved a low average percentage of 0.21% in terms of "Ramachandran outliers", a high percentage of 93.89% in terms of "Ramachandran favored", and a low average value of 2.11 in terms of "MolProbity score" on the test set of 47 single-particle EM maps (Supplementary Data 4a). Similar quality can also be observed in the EMBuild models with an average value of 0.17% for "Ramachandran outliers", 93.81% for "Ramachandran favored", and 2.27 for "MolProbity score" on the test set of 16 subtomogram averaging EM maps (Supplementary Data 4b).

**Evaluating EMBuild models against higher resolution structures.** We have evaluated the accuracy of built models using the deposited PDB structure as the ground truth reference. However, the PDB structure might contain errors due to the low resolution of its associated EM map, even though the associated PDB models are generally optimized into the deposited map by the depositors. Thus, we added two extra test cases for EMBuild using the higher resolution structures as the reference. Namely, the EMBuild models were built on intermediate-resolution EM maps and then evaluated against the higher resolution reference structures. The first test case is EMD-2788 of horse spleen apoferritin at a resolution of 4.7 Å. EMBuild built a perfect model on this intermediate resolution map, and achieved a TM-score of 0.999 and an RMSD of 0.58 Å with respect to the 1.5 Å reference crystal structure (Supplementary Fig. 1a). The other test case is EMD-12661, which is a 2.1 Å cryo-EM map for respiratory complex I. Given its high resolution, the accuracy of the associated PDB model could be ensured. Then, an 8.0 Å map was low-pass filtered from the half-maps of EMD-12661 using RELION post-

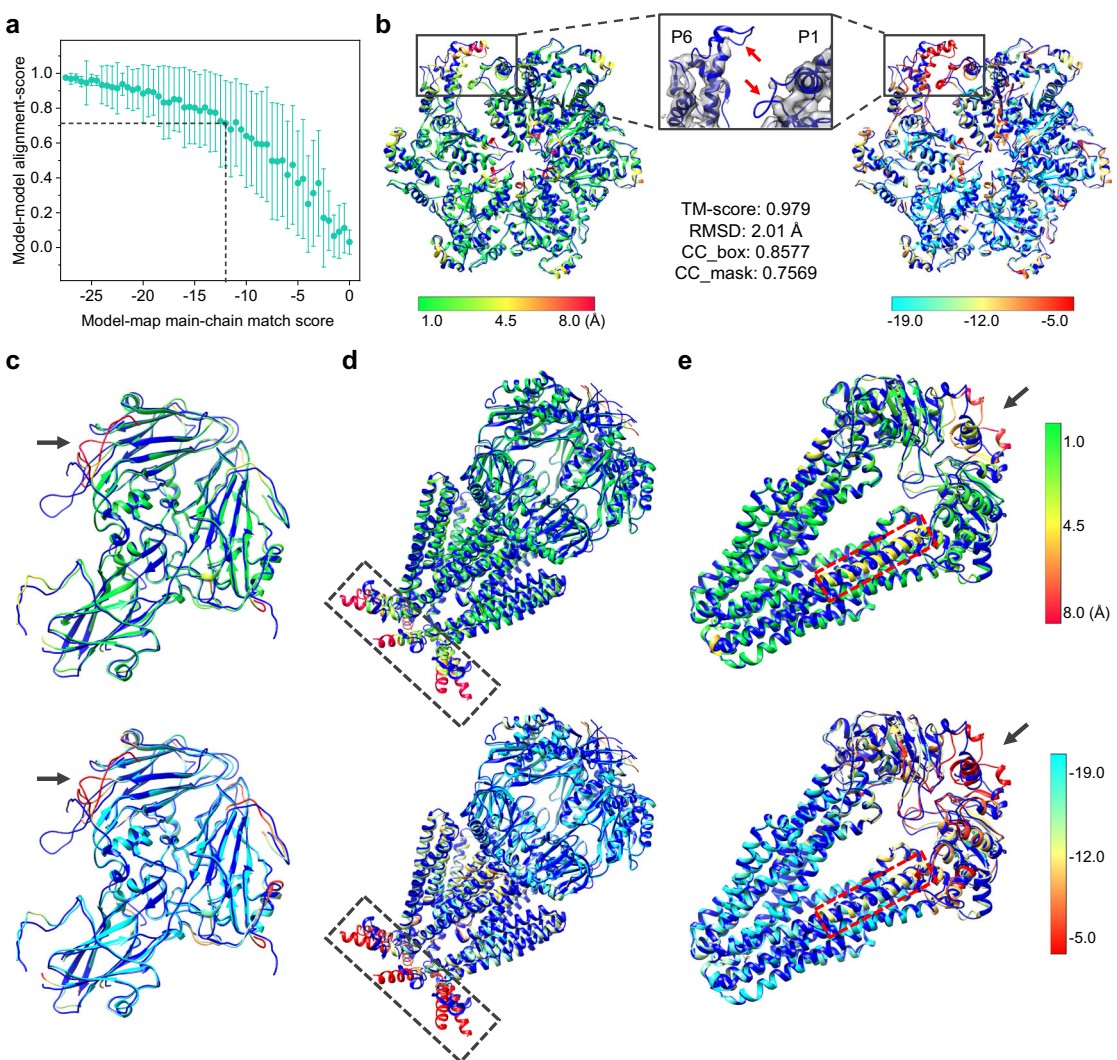

**Fig. 7 Quality assessment of the built models by EMBuild. a** Main-chain match scores versus alignment scores of $n = 7308$ continuous secondary structure fragments with no less than five residues, where the match scores are merged into bins of size 0.5. The alignment scores in each bin are presented as mean values ±1.0 standard deviation. Exact number of data points in each bin is provided in the Source Data file. **b** The color-scaled display of the built complex model by EMBuild in terms of Cα displacements (left) and main-chain match scores (right) of each residue for EMD-8794 (PDB ID: 5WC0). The enlarged view in the middle displays the density volume between the first and last protomers (P1 and P6). The reference PDB structures are colored in blue. Red arrows indicate two low-density regions on P6 and P1, respectively. **c–e** The color-scaled display of the built complex model by EMBuild in terms of Cα displacements (top) and main-chain match scores (bottom) of each residue for EMD-9631 (PDB ID: 6AJ2) (**c**), EMD-20510 (PDB ID: 6PWP) (**d**), and EMD-20950 (PDB ID: 6UZ2) (**e**), respectively. Regions that show differences between the predicted model and the PDB structure are indicated by arrows or boxes, which are accurately identified by the match scores. Source data are provided in the Source Data file.

processing[60]. As shown in Supplementary Fig. 1b the EMBuild model recovered the conformation of the high-resolution reference structure on the 8.0 Å map, and yielded a TM-score of 0.986 and an RMSD of 1.69 Å.

**Impact of anisotropy in EM maps.** Intermediate-resolution maps may suffer from preferred orientation and resulting anisotropy in density signals. Therefore, we examine the impact of anisotropy in EM maps on EMBuild. Our results revealed that EMBuild maintained good performance in such scenario, which is attributed to two aspects of our method. On one hand, by converting the input density map to a main-chain probability map through deep learning, the anisotropy in EM density map can be to some extent mitigated by EMBuild. As shown in Fig. 8a–c, the density map of EMD-20501 exhibits both angular anisotropy and radial anisotropy[61], which are absent in the corresponding main-chain probability map (Fig. 8d–f). On the other

hand, the impact of anisotropy can be alleviated by the structural context of the AlphaFold2-predicted structure. As shown in Fig. 8g, h, the untilted reconstruction for Influenza hemagglutinin (HA) trimer suffers from severe anisotropy that causes missing density in the map[62]. Nevertheless, with the accurate structures of chains predicted by AlphaFold2, EMBuild built a rational complex model on such an extremely low-quality map with a TM-score of 0.954 and an RMSD of 2.78 Å. The EMBuild model conforms the PDB structure well on regions with sufficient density signals, while those regions that are not optimally modeled due to missing density can also be identified by main-chain match scores for further improvement.

**Impact of protein symmetry.** Although symmetry is ignored during the evaluations for general applicability, protein symmetry is valuable information in the model building of EM maps. Therefore, we also evaluated whether symmetry information can

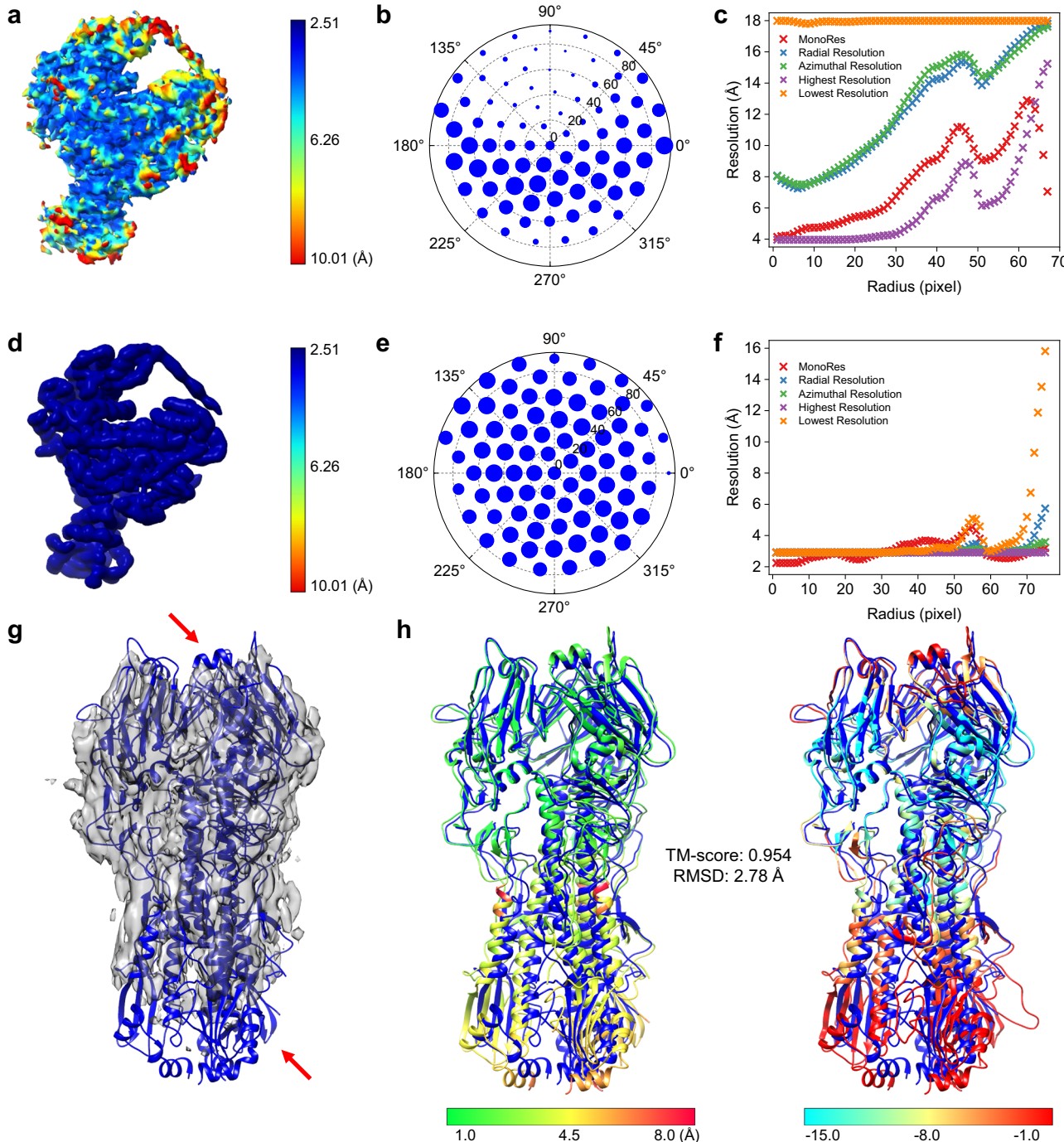

**Fig. 8 Model building by EMBuild on an EM map with anisotropy. a–f** Comparison of local anisotropy for the EM density map (**a–c**) and the main-chain probability map (**d–f**) of EMD-20501, respectively. **a**, **d** Local resolution map. **b**, **e** Angular plot of local-directional resolution map. **c**, **f** Radial average of local-directional resolution map. **g**, **h** The EMBuild model built on an untilted reconstruction of Influenza hemagglutinin (HA) trimer (EMPIAR-10096) that has severe anisotropy in the map. **g** The reference PDB model (PDB ID: 7VDF) colored in blue superimposed onto the EM map colored in transparent gray. Red arrows indicate the regions with missing density. **h** Comparison between Cα displacements and main-chain match scores of the EMBuild model. The EMBuild model is colored by Cα displacements to the deposited PDB structure (left) and is colored by main-chain match scores (right). The reference deposited PDB structure is colored in blue.

help to improve the performance of EMBuild on a subset of 19 maps that have C or D symmetry out of the 47 single-particle EM maps. For each map, symmetry matrices are calculated on the density maps by phenix.map_symmetry. To ensure that each map can find its symmetry center, the symmetry center and symmetry type of the deposited PDB structure is provided to phenix.map_symmetry. The comparison between the performance of EMBuild with and without using symmetry information is shown in Fig. 9a–c. The detailed evaluation results for each of the test cases are listed in Supplementary Data 5. It can be seen from the figure that the performance of EMBuild is not significantly changed after applying symmetry information on most of the test cases, which demonstrated the capability of EMBuild in recovering the symmetry of protein complex without using symmetry

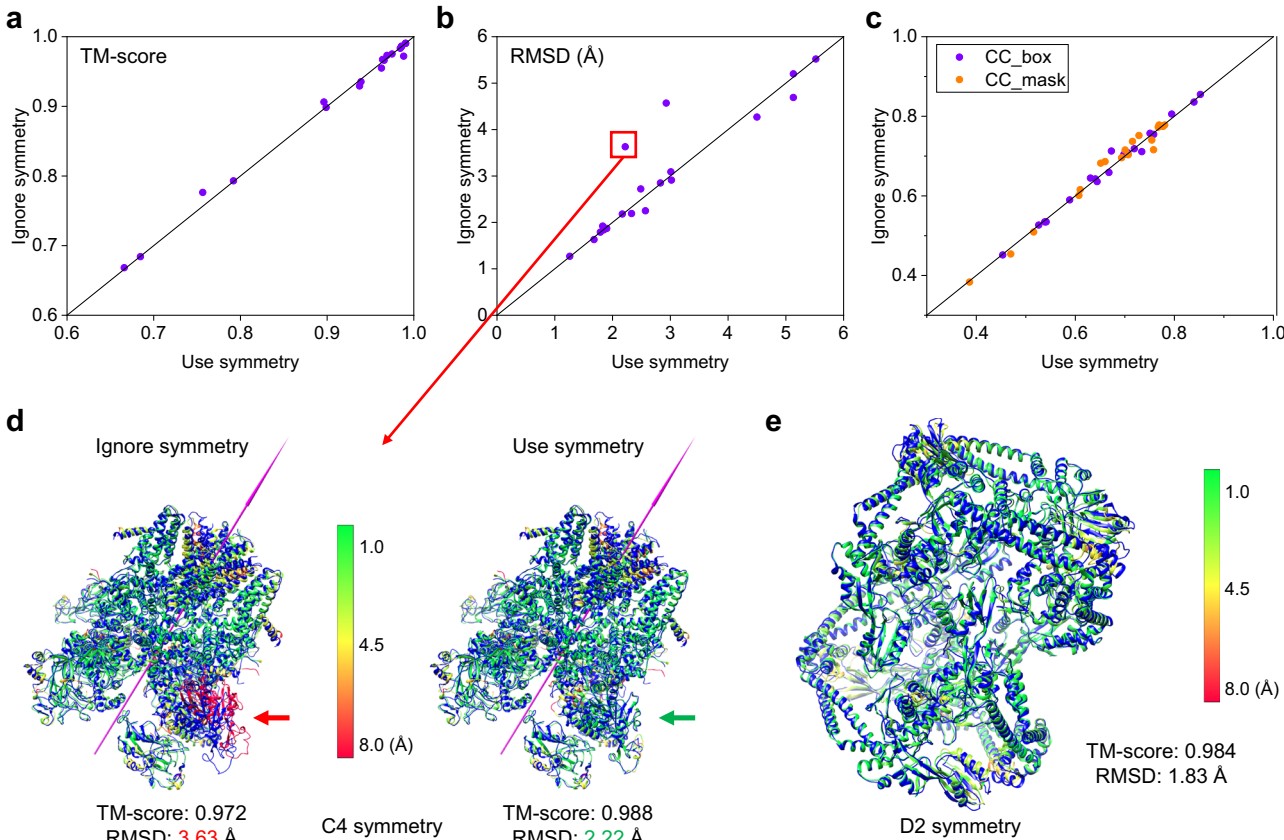

**Fig. 9 Comparison of the EMBuild models with and without using symmetry information, on the test set of 19 single-particle EM maps with C or D symmetry. a–c** Comparison of TM-score (**a**), RMSD (**b**), and CC values (**c**) between the EMBuild models built with and without symmetry information on each test case. Highlighted in the red box of **b** is the case that has a significantly lower RMSD value after using symmetry information. **d** The example of EMD-20479 highlighted in **b** with C4 symmetry. The deposited structures are colored in blue, and the EMBuild models built with symmetry information (left) and without symmetry information (right) are colored by Cα displacements to the deposited PDB structure. Arrows colored in magenta indicate the axis of symmetry. Indicated by red and green arrows is a part of structure that is improved after using symmetry information. **e** An example with D2 symmetry (EMD-9317). The deposited PDB structure is colored in blue, and the EMBuild model with symmetry information is colored by Cα displacements to the deposited PDB structure. Source data are provided in the Source Data file.

information. Nevertheless, EMBuild can indeed benefit from symmetry information in some cases. Taking EMD-20479 with C4 symmetry as an example, a domain in one of the four protein chains was poorly modeled without using the symmetry information, resulting in a high RMSD value of 3.63 Å (Fig. 9d). However, after applying the symmetry information to EMBuild, the quality of EMBuild model can be improved and the RMSD value was improved to 2.22 Å. Besides C symmetry, EMBuild is capable of dealing with D symmetry. For example, EMBuild built a good model of D2 symmetry on EMD-9317, as shown in Fig. 9e. The build model achieved a TM-score of 0.984 and an RMSD of 1.83 Å.

**Impact of input structure trimming**. During the evaluations, we trimmed the AlphaFold2-predicted structure according to the PDB structure for ease of comparison. However, the starting and ending residues in the structure may not be known in real applications. Therefore, we also evaluated the performance of EMBuild using a more objective trimming protocol, that is, trimming the structure according to the pLDDT values predicted by AlphaFold2. Specifically, continuous fragments of residues with pLDDT value < 50 at the N- and C-terminals are removed from the predicted structure. To avoid severe inconsistency between the structure and the sequence, cases that have any chain with <50% coverage of the full-length sequence are excluded,

yielding a subset of 34 cases out of 47 single-particle EM maps. A comparison of EMBuild with PDB structure trimming and pLDDT trimming is shown in Supplementary Fig. 2. The detailed results are listed in Supplementary Data 6. It can be seen from the figure that the performance of EMBuild with PDB structure trimming shows no significant difference from the case of realistic pLDDT trimming, in terms of TM-score and RMSD values. The average TM-score and RMSD achieved by EMBuild with pLDDT trimming are 0.904 and 2.96 Å, respectively, which are about the same as 0.903 and 2.94 Å for PDB structure trimming.

**Applying main-chain probability maps to other methods**. One important strategy in EMBuild is the use of the main-chain probability maps. To investigate whether such main-chain probability map can help model building of other approaches, we also applied the main-chain probability maps to phenix.dock_in_map and gmfit on the test set of 47 single-particle EM maps. Specifically, for each input protein chain, only main-chain atoms (N, C, and Cα) are kept during fitting into the main-chain probability map, and the rest atoms are ignored until the assembling is finished. The comparisons for model building on the raw EM density map and main-chain probability map are shown in Supplementary Fig. 3. The detailed results are listed in Supplementary Data 2. It can be seen from the figure that the performance of phenix.dock_in_map was significantly improved

on the main-chain probability maps. The average TM-score, RMSD, CC_box, and CC_mask values achieved by phenix.dock_in_map on the main-chain probability map are 0.807, 4.03 Å, 0.6265, and 0.5686, compared to 0.746, 4.52 Å, 0.5975, and 0.5323 on the original EM density map. The improvement of phenix.dock_in_map on the main-chain probability maps demonstrated the importance of the main-chain probability maps in model building. However, gmfit did not benefit from main-chain probability maps. This is understandable because gmfit converts the input map into a Gaussian mixture model in the fitting procedure, which may miss detailed structural information in the main-chain probability maps.

**Capability of EMBuild in identifying native components.** In the evaluations, we have built the complex model from its native sequences. However, we may not know with certainty which components are present in a given map in some situations[63]. Therefore, we also evaluated the capability of EMBuild to identify the native sequence for a given cryo-EM map from a pool of candidate sequences. Here, we take EMD-0290 as an example, which is a homotrimer solved by tomography averaging of cryo-ET data at 7.2 Å resolution. The native sequence of one component has 200 residues. To build a pool of decoy sequences, 298 diverse sequences of length 200 are cropped from all the unique sequences in the two test sets. Then, the AlphaFold2 structures for 298 decoy sequences plus one native sequence are fitted into the map using EMBuild. The sequences that failed to form a trimer structure in the map are excluded. The mean main-chain match score of three chains is calculated. As shown in Supplementary Fig. 4, EMBuild is able to distinguish the native sequence from decoy sequences based on the main-chain match score. The native sequence has the best main-chain match score of −11.06 among the pool of candidate sequences (Supplementary Data 7). The success of EMBuild here can be partially attributed to the fact that all of the decoy sequences have a low sequence identity of <25% of the native sequence. To distinguish between homologous sequences that have similar main-chain structures, additional experiments would be needed to verify the native sequence.

**Computational efficiency.** The running time of EMBuild comes from two parts. One is to predict the main-chain probability map from the EM density map, and the other is to assemble the complex structure using the main program. The detailed running times on the test set of 47 single-particle EM maps are listed in Supplementary Data 8. It can be seen from the table that the prediction of the main-chain probability map can normally be finished within 100 seconds on four NVIDIA A100 GPUs. As for the main program of EMBuild, it consumes an average time of 3838.2 seconds on a single core of Intel(R) Xeon(R) Gold 6240 for each test case, compared with 1585.7 seconds for phenix.dock_in_map. The longer computing time of EMBuild than phenix.dock_in_map is understandable because EMBuild includes more time-consuming semi-flexible domain refinement in addition to rigid fitting. Nevertheless, by parallel computing, the running time of EMBuild can be drastically reduced. For example, the running time is reduced to 291.4 seconds by parallelly running EMBuild on 36 cores using OpenMP. Compared to EMBuild and phenix.dock_in_map, gmfit consumed much more time with an average of 15514.3 seconds for each case because an extremely large number of configurations are searched and refined in gmfit (Supplementary Data 8).

## Discussion

In this study, we have proposed EMBuild, an automatic model building method for intermediate-resolution cryo-EM maps

through iterative fitting and refinement of protein fragments. Due to its semi-flexible fitting capability, EMBuild is able to build accurate protein complex models into EM maps as long as the predicted protein structures of individual chains are reliable at fragment or domain level. Compared with traditional rigid-fitting methods, EMBuild is especially robust and achieves the most improvement for those cases where the proteins experience global conformational changes through the relative movement of rigid domains/fragments. For those cases where the AlphaFol2-predicted structure is not accurate, EMBuild tends to perform less satisfactorily. However, such limitations can be much alleviated by removing those inaccurate or intrinsically disordered parts based on their pLDDT values in the AlphaFold2-predicted structure. In addition, the performance of EMBuild mainly depends on the resolution of cryo-EM maps. As such, EMBuild can perform equally well for those samples prepared by different techniques like continuous carbon coating as long as their reconstructed 3D maps have similar resolutions.

It should also be noted that since primary maps are used for training our main-chain prediction model, sharpened maps are recommended as the input of EMBuild for the best performance of main-chain probability prediction. As shown in Supplementary Fig. 5, the sharpened map by RELION[60] with an automatically determined B-factor can significantly improve the quality of the predicted main-chain probability map, compared with the unsharpened half-maps.

In summary, we have developed a deep learning-guided method for automatic model building of protein complex structures from intermediate-resolution cryo-EM maps, which is referred to as EMBuild. EMBuild was extensively evaluated on diverse test sets of 47 single-particle maps and 16 subtomogram averaging maps, and compared with existing approaches including phenix.dock_in_map, DEMO-EM, and gmfit. It was shown that EMBuild outperformed the other methods and was able to build reliable atomic models with an average TM-score of 0.909 and an RMSD of 2.85 Å with respect to the PDB structures on the test set of single-particle maps, and 0.863 and 2.74 Å on the test set of subtomogram averaging maps, respectively. In addition, the built models by EMBuild also showed a competitively high quality to the manually built PDB structures in the validation against the EM map density. Furthermore, EMBuild also provides a reliable way to assess the quality of the built model, enabling the accurate interpretation of the built model. Combined with accurate protein structure prediction techniques, EMBuild is able to automatically build high-quality protein complex models for any intermediate-resolution cryo-EM maps. The human-level accuracy of the built model by EMBuild would make the model ready to be used after minor adjustments. It is anticipated that EMBuild will serve as an indispensable tool for streamlining the structure determination from intermediate-resolution cryo-EM maps.

## Methods

**Data collection**. We have collected a non-redundant dataset of cryo-EM maps from the EMDB. The primary maps, which are normally sharpened maps, are used in this study.All the single-particle EM entries at 4–8 Å resolution that have associated PDB models were downloaded from the EMDB[17] and PDB[18]. Specifically, the PDB structure for each entry was directly downloaded from the PDB at https://www.rcsb.org/, and its corresponding density volume within 4.0 Å of the PDB structure was segmented out from the whole map. The EM map and its corresponding PDB structure that have any of the following features were removed: (i) without side-chain atoms, (ii) including unknown residues (UNK), (iii) including missing chain or non-protein chain, (iv) having nonorthogonal map axis, (v) corresponding to multiple PDB or EMDB entries, and (vi) having severe misfits between the PDB model and EM map. In order to ensure the feature (vi), we calculated the cross-correlation between the deposited map and the map simulated from the PDB model at the same resolution using the UCSF chimera[30]. Any map and its associated PDB model that have a cross-correlation of less than 0.65 were

excluded. Afterwards, all the remaining maps were manually checked. The remaining cases were retained as the initial dataset. To remove redundancy, the initial dataset of cases was clustered using greedy algorithm. Two models are considered to be similar if any chain in the first model has >30% sequence identity with any chain in the second model. The one with the largest number of similar cases is chosen as the representative of the corresponding cluster, and then the rest cases in the cluster are removed. This procedure is repeated until all the cases are clustered. The resulted non-redundant training set consists of the representatives of each cluster. A total of 262 pairs of EM maps and associated PDB structures were retained. In order to train a deep learning model for predicting main-chain probability map, 209 maps were randomly selected as the final training set (Supplementary Data 1).

To build an independent test set with a sufficient number of valuable cases, all the cases in the initial dataset that have >30% sequence identity with any case in the training set are excluded. Then, any pair of EM map and its associated PDB structure that meet the following criteria are removed: (i) having only one chain and (ii) having >30% gap in the structure according to the gene sequence. For saving time in evaluations, we have also removed those cases with more than 10 chains, though our method can work with any number of chains. The remaining cases are clustered using a sequence identity cutoff of 70% by the similar greedy algorithm described above. The final test set contains 47 single-particle cryo-EM maps with resolutions ranging from 4.0 to 8.0 Å (Supplementary Data 2). In the modeling and evaluating processes, short chains that have less than 20 residues are ignored.

To evaluate the performance of EMBuild on the maps by subtomogram averaging of cryo-ET data, we further constructed another independent test set of subtomogram averaging maps. All the subtomogram averaging maps with resolutions within 10.0 Å that have associated PDB models are filtered and clustered using the same way as that for the test set of single-particle maps, except for allowing more than 10 chains. The final test set contains 16 subtomogram averaging maps with resolutions ranging from 3.7 to 9.3 Å (Supplementary Data 3).

**Main-chain probability prediction by deep learning**. We used a deep learning model to predict the main-chain probability map from the input raw EM density map through a nested U-net (UNet++)[51]. The network architecture consists of three encoder sub-blocks and three decoder sub-blocks with dense skip connections, where 3D convolution layers with a kernel size of $3 \times 3 \times 3$ are applied (Fig. 1a). The 3D maxpooling layer with stride of 2 is adopted for down-sampling, and the trilinear interpolation layer with zoom factor of 2 is adopted for up-sampling. During the prediction, the raw EM map is rescaled to have a grid interval of 1.0 Å by applying cubic interpolation and cut into overlapping chunks of size $40 \times 40 \times 40$ with slide strides of 10 voxels, which are input into the deep learning model. Then, the predicted main-chain probability chunks with the same size and grid interval are re-assembled into the final main-chain probability map by averaging overlapping parts.

A set of pairs of experimental EM density and main-chain probability maps were used to train the deep learning model. For the EM maps, the grid interval is rescaled to 1.0 Å by applying cubic interpolation. The density values are clipped to be equal or greater than 0.0 and are normalized to the range 0.0–1.0 by the 99.999-percentile density value of the map. For a given EM density map, its main-chain probability map is generated from its associated PDB structure with 1.0 Å grid interval, where the main-chain probability $p$ for a grid point $\mathbf{x}$ is calculated as follows

$$p(\mathbf{x}) = \max\{e^{-\lambda\|\mathbf{x}-\mathbf{a}\|^2}, \forall \mathbf{a} \in A\} \quad (1)$$

where $A$ stands for the set of position vectors of all main-chain atoms (N, C, and Cα). The value of $\lambda$ is defined as

$$\lambda = (\pi/(2.4 + 0.8R))^2 \quad (2)$$

where $R$ is the map resolution[43]. For training, the EM density maps and their corresponding main-chain probability maps are cut into pairs of overlapping boxes of size $60 \times 60 \times 60$ with slide strides of 30 voxels, where non-positive boxes are excluded for effective training.

During the training, 20% of the maps are randomly selected from the training set as the validation set. The training data are augmented through random 90° rotations and randomly cropping a $40 \times 40 \times 40$ chunk from the input $60 \times 60 \times 60$ box. The network is implemented in Python with Pytorch1.8.1 + cuda11.1. For each model, the network is trained for at most 300 epochs with 160 boxes employed in one batch. The Adam optimizer is adopted to minimize the loss of the prediction, where the total loss is a sum of two different loss functions. One is the smooth L1 loss, which calculates the numerical difference in the probability values between the predicted chunk $X$ and target chunk $Y$ as follows

$$\text{SmoothL1Loss}(X, Y) = \sum_{i=1}^{N}\sum_{j=1}^{N}\sum_{k=1}^{N}\frac{L_{i,j,k}}{N^3} \quad (3)$$

where $N$ is the chunk size, and $L_{i,j,k}$ is the Smooth L1 distance between $X$ and $Y$ at position $(i, j, k)$. The other is the structural similarity (SSIM) loss which compares the contrast and structure similarity between a predicted chunk $X$ with its target

chunk $Y$ according to the following formula

$$\text{SSIMLoss}(X, Y) = 1 - \frac{2\sigma_{XY} + \varepsilon}{\sigma_X^2 + \sigma_Y^2 + \varepsilon} \quad (4)$$

where $\sigma_X$ and $\sigma_Y$ are the standard deviations for the predicted chunk $X$ and target chunk $Y$, $\sigma_{XY}$ is the covariance between $X$ and $Y$, and $\varepsilon$ is set to be a small constant ($\varepsilon = 10^{-6}$ in this study) to prevent dividing by zero. The learning rate is initially set to $10^{-3}$ and will be reduced to 1/2 of its current value if the average loss on the training set does not decrease for 4 epochs. The training process continues until the learning rate reaches a minimum value of $10^{-5}$. The network model with the least validation loss is selected. As seen from the learning curves, the training and validating losses converge well for our deep learning model (Supplementary Fig. 6).

**Fitting protein chains into the main-chain probability map**. The structure models of individual chains can be predicted from their sequences by a protein structure prediction program. In this study, AlphaFold2 was selected for this purpose, given its excellent performance in protein structure prediction[52]. To mimic real situations, the corresponding PDB structure and those newer structures are excluded from the templates by setting the "max template data" to the day before the released date of the corresponding PDB structure[52], and full-length gene sequences are used as the input. For consistency, the predicted model of each chain is cropped to have identical starting and ending residues with the sequence in the PDB structure. It should be noted that during the implementation of EMBuild there is no structural gap and/or uninterpreted region in our predicted chain models, while they may possibly present in the PDB structure. For each chain model predicted by AlphaFold2, we use SWORD to assign its structural domains[64]. SWORD will generate multiple assignments for one chain. The assignment with the most domains with no less than 30 residues and a $\kappa$ value of no less than 3.0 is chosen as the final domain assignment.

For computational efficiency, instead of directly using the predicted main-chain probability map, a reduced representation of main-chain probabilities is adopted by EMBuild. Namely, a mean shift algorithm is applied to generate representative points of main-chain probabilities. Specifically, starting from the positions of positive grid points on the main-chain probability map, the seed points $\mathbf{z}_i^t (i = 1, ..., N'; t = 0, 1, ...)$ is iteratively shifted to local maxima of probabilities as

$$\mathbf{z}_i^{t+1} = \frac{\sum_{n=1}^{N} K(\mathbf{z}_i^t - \mathbf{x}_n)p(\mathbf{x}_n)\mathbf{x}_n}{\sum_{n'=1}^{N} K(\mathbf{z}_i^t - \mathbf{x}_{n'})p(\mathbf{x}_{n'})} \quad (5)$$

where $\mathbf{x}_n (n = 1, ..., N)$ are the position vectors of grid points, $K(\mathbf{z}_i^t - \mathbf{x}_n)$ is the Gaussian kernel function, and $p(\mathbf{x}_n)$ is the main-chain probability of grid point $\mathbf{x}_n$. The Gaussian kernel function is described as $K(\mathbf{z}_i^t - \mathbf{x}_n) = e^{-\lambda\|\mathbf{z}_i^t - \mathbf{x}_n\|^2}$. The main-chain probability of a shifted seed point $P(\mathbf{z}_i^t)$ is computed as $P(\mathbf{z}_i^t) = \frac{1}{N}\sum_{n=1}^{N} K(\mathbf{z}_i^t - \mathbf{x}_n)p(\mathbf{x}_n)$. After the mean shift procedure is converged, the seed points that are closer than a threshold distance are clustered and the one with the highest probability value is chosen as the representative of each cluster. The resulted points $\mathbf{z}_i (i = 1, ..., L) \in Z$ are referred to as the main-chain points.

We adopt a fast Fourier transform (FFT)-based matching strategy to globally fitting the protein model for each chain to the main-chain probability map (main-chain points)[65]. To perform an exhaustive FFT-based search, both the protein model and main-chain points are first mapped onto a three-dimensional (3D) grid of shape $M \times M \times M$ with 1.5 Å grid interval. The main-chain probabilities are assigned to the grid of the main-chain points (say grid $A$) and the grid of the main-chain atoms in the protein model (say grid $B$), according to Eq. (1). With the above main-chain probability mapped on grids, the match score $S$ for a superimposition between the protein model and main-chain probability map can be generally expressed by the following formula

$$S(i, j, k) = -\theta \sum_{l=1}^{M}\sum_{m=1}^{M}\sum_{n=1}^{M}(A_{l,m,n} \times B_{l+i,m+j,n+k}) \quad (6)$$

where $i$, $j$, and $k$ are the numbers of grid points by which the protein model is shifted with respect to the main-chain points in three translational dimensions, and $\theta$ is the resolution-dependent factor defined as $\theta = (\lambda/\pi)^{1.5}$. The match scores for all the $M^3$ translations can be computed through one round of FFT-based calculation. The rotational search is conducted by exploring a large set of rotation angles. That is, for each rotation of the protein structure, an FFT-based translational search is carried out. For EMBuild, an angle interval of 15° is used to evenly discretize the Euler space, which results in a total of 4392 evenly distributed orientations. The fitting results of the exhaustive search are further optimized through a SIMPLEX method. The match score $s'$ of main-chain atom $\mathbf{y}_q (q = 1, ..., Q)$ in protein pose $Y$ can be calculated as

$$s'(\mathbf{y}_q; Z) = -\max\{\theta P(\mathbf{z})e^{-\lambda\|\mathbf{y}_q - \mathbf{z}\|^2}, \forall \mathbf{z} \in Z\} \quad (7)$$

where $\mathbf{z} \in Z$ is the position of the main-chain point and $P(\mathbf{z})$ is its main-chain probability. The match score for pose $Y$ is the summation of scores of included atoms as $S'(Y; Z) = \sum_q^Q s'(\mathbf{y}_q; Z)$. Finally, the fitting results are ranked by the match scores and those top-scored poses are retained.

After the protein model of a chain is rigidly fitted to the main-chain probability map, we employed a semi-flexible domain refinement strategy to further improve

the fitness between the protein models the map, as illustrated in Fig. 2a. For each protein model, n short structure domains are assigned by SWORD[64]. A simple graph is built based on the domain assignment, where two domains with connecting residues are connected by an undirected edge. The domain refinement is applied on each of the $M$ top-scored poses from rigid fitting. Starting from a selected domain as the seed domain, the positions of all domains are optimized one after another. Specifically, starting from the seed domain, a SIMPLEX optimization is conducted to find a locally best match of the current domain, and then the optimization is carried out to the neighboring domain of current domain. This procedure is repeated until all the domains are locally optimized. The order of the refinement is determined by breadth-first search (BFS) on the domain graph. Taking each domain as the seed domain, the domain refinement will output $n$ different models. The results of domain refinement for one protein chain are $M \times n$ refined models plus $M$ rigidly fitted models, which are ranked by their match scores.

**Assembling chains into protein complex.** After fitting and refinement of individual chains, the last step of EMBuild is to assemble individual chains into a reliable complex structure. To prevent clash between different chains, we defined a clash score $c$ of one atom $\mathbf{a}$ of chain $A$ with respect to all atoms $\mathbf{b}$ of another chain $B$ as follows

$$c(\mathbf{a}; B) = \max\{e^{-\lambda(\max\{\|\mathbf{a}-\mathbf{b}\|-d_{clash}, 0.0\})^2}, \forall \mathbf{b} \in B\} \quad (8)$$

where $d_{clash}$ stands for the cutoff distance within which the clash score is set to 1.0. The clash score $C$ of chain $A$ with respect to chain $B$ is the average clash score of all $n_A$ atoms with $C(A; B) = \frac{1}{n_A} \sum_{\mathbf{a} \in A} c(\mathbf{a}; B)$. According to the match scores of individual chains and the clash scores between different chains, the problem of assembling chains into a complex will become a Maximum Clique Problem. First, an undirected graph is built by EMBuild, of which the vertices are the match scores of individual chains, and the edges connecting two different chains are the corresponding clash scores. The different fitting poses of one chain are not connected. Then, the edges that have a clash score exceeding a given threshold $C_{thr}$ are removed from the graph. After the graph is built, the Bron-Kerbosch algorithm is used to find the best combination of chains from the graph that has the highest total match score. However, though only a single cycle of Bron–Kerbosch algorithm, some chains may not be assembled into the protein complex. Therefore, we adopt an iterative strategy in EMBuild, as illustrated in Fig. 2b. Namely, after each round of the Bron-Kerbosch algorithm, the clash scores of main-chain points are calculated according to the current assembled complex structure $D$, which are used to update the probability $P(\mathbf{z}_i)(i = 1, \ldots, L)$ of main-chain points, according to the following formula

$$P'(\mathbf{z}_i; D) = P(\mathbf{z}_i) \times (1.0 - c(\mathbf{z}_i; D)) \quad (9)$$

By updating the probabilities of main-chain points, the regions with fitted structures are removed from further assembling. The remaining chains are iteratively assembled to the complex structure by fitting and refining in accordance with the updated main-chain points. Finally, the resulted complex model is refined in the EM density map using phenix.real_space_refine[47].

**Evaluation metrics.** Three types of metrics are used to evaluate the quality of the protein complex model built by EMBuild. The first type of metrics is one that measure the closeness between the built model and the PDB structure. In this respect, we adopt the TM-scores and RMSDs between the built complex model and the corresponding PDB structure calculated by MM-align[54,55]. The second type of metrics is those fit-to-map metrics of the built models[56], which measure the consistency between the built model and the EM density map. In this regard, we report CC_box and CC_mask values calculated by phenix.map_model_cc and the map-model FSC05 calculated by phenix.mtriage[57]. It is noted that phenix.mtriage will fail to give a valid FSC05 value if the built model does not conform to the map, where the maximum value of the map-model FSC is only around or even below 0.5 over the entire resolution range. Besides the above metrics, we also reported coordinates-only metrics including the Ramachandran scores and MolProbity score calculated by MolProbity[59].

**Quality assessment of built models.** To assess the quality of the model built by EMBuild, we propose a metric of the main-chain match score, which measures the fitness between the built model and the main-chain probability map. The match scores for individual main-chain atoms are calculated according to Eq. (7). Starting from the match scores of main-chain atoms, it is easy to calculate the average score of three main-chain atoms as the match score for each residue. By analogy, we further calculate the match score of each continuous secondary structure fragment as the average match score of its containing residues. The secondary structure is assigned by STRIDE[66]. The final match score of a fragment is defined as a combination of its initial score and the match score of its domain as

$$S'_{fragment} = 0.7 \times S_{fragment} + 0.3 \times S_{domain} \quad (10)$$

where $S_{domain}$ is the average match score of all residues in the domain that contains the fragment. Similar to the fragment match score, the final match score of a

residue is defined as a combination of its initial score and the match score of its fragment as

$$S'_{residue} = 0.7 \times S_{residue} + 0.3 \times S_{fragment} \quad (11)$$

The resulted residue scores are further smoothed along the chain with a sliding window of weights 1:2:4:8:16:8:4:2:1 centered at each residue.

To measure the quality of modeled fragments with respect to the PDB structure, we define an alignment score, which is described as

$$\text{alignment} - \text{score} = \frac{1}{L} \sum_{i=1}^{L} \frac{1}{1 + d_i^2/d_0^2} \quad (12)$$

where $L$ is the length of the fragment, $d_i$ is the distance of the $i$th pair of the aligned residues between the built model and the fragment in the PDB structure, and $d_0$ is a scale factor. Although our alignment score takes a similar expression to TM-score[55], two significant differences should be noticed. One is that no superposition is applied before calculating the alignment score. The other is that $d_0$ is set to a fixed distance of 3.0 Å.

**Comparison with related methods.** EMBuild is compared with phenix.dock_in_map, DEMO-EM, and gmfit on the test sets of cryo-EM maps. For each test case, the protein models of individual chains are predicted from sequences by AlphaFold2, which are used as the input for different methods. To be general, the symmetry information of test cases is ignored during the evaluation. phenix.dock_in_map uses both the secondary structure matching and convolution-based shape searches to find a part of a map that is similar to a protein model, which can be used to place any number of copies of any number of unique molecules[42]. gmfit is a program for fitting subunits into a density map using GMM (Gaussian Mixture Model)[31,32]. To convert the density maps and protein chains into a GMM used by gmfit, the number of Gaussian functions for a density map is set to 20 multiplied by the number of chains in its associated protein complex. The number of Gaussian functions for a query protein chain is set to 20. For gmfit, the number of randomly generated initial configurations is set to 100000, the number of configurations for search is set to 20000, and the number of configurations for refinement is set to 4000. DEMO-EM is a hierarchical method to assemble multi-domain protein structures from cryo-EM density maps[53]. We have tested DEMO-EM on the test set of single-particle maps in two ways. One is inputting protein sequences, where structure predictions are carried out by DEMO-EM itself. The other is providing structures predicted by AlphaFold2 to DEMO-EM. It should be noted that 22 cases with a total sequence length >2000 are not accepted by the DEMO-EM server. Thus, the rest 25 cases were submitted to the DEMO-EM online server, of which EMD-22216 failed using AlphaFold2 structures as the input. For every model building method, the resulted complex model is refined in the EM density map using phenix.real_space_refine[47].

**Reporting summary.** Further information on research design is available in the Nature Research Reporting Summary linked to this article.

## Data availability
Data that support the findings of this study are available from the corresponding author upon request. The Source Data underlying Figs. 3, 5, 6a–d, 7a, 8b, e, c, f, 9 and Supplementary Figs. 3, 4, 6 are provided as a Source Data file. All published data sets used in this paper were taken from the EMDB and PDB (accession codes specified in the figure captions and the tables in Supplementary Data).

## Code availability
The EMBuild package is freely available for academic or non-commercial users via http://huanglab.phys.hust.edu.cn/EMBuild/.

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

## Acknowledgements

This work was supported by grants from the National Natural Science Foundation of China to S.H. (32161133002 and 62072199) and the startup grant of Huazhong University of Science and Technology.

## Author contributions

S.H. conceived and supervised the project. J.H. and S.H. designed and performed the experiments. P.L. generated AlphaFold2 models. J.C. benchmarked gmfit. H.C. evaluated the software, J.H. and S.H. wrote the manuscript. All authors read and approved the final version of the manuscript.

## Competing interests

The authors declare no competing interests.
