## [Peer Review File · Nature Communications]

Model building of protein complexes from intermediate-resolution cryo-EM maps with deep learning-guided automatic assemblyReviewers' Comments:

Reviewer #1:

Remarks to the Author:

The authors have developed a program to build protein models into intermediate (4-8 Å) resolution EM maps for complexes made up of 2 – 10 protein components. Their approach is innovative in that it combines and builds on recent advances in model building. First, rather than modeling directly into the map, the map is used to predict a main-chain probability map using nested UNet++. Second, AlphaFold2 or other external programs are used to predict the 3D structure of the components of the protein complex of interest based on sequence. Finally, the sequence-based structural models are iteratively fit and refined in segments into the main-chain probability map. The authors trained the program on over 200 unique pairs of maps from the EMDB with corresponding models in the PDB. They then applied this approach, as well as other modeling programs, to 47 additional EMDB map entries that have corresponding PDB entries. The authors found that their approach performs as well as or better than other modeling approaches at producing results similar to the structures deposited in the PDB from the corresponding EMDB maps.

Building models into intermediate resolution EM maps is an important and significant challenge in the field. Because there is such great variation in biological samples, additional tools to address this challenge in innovative ways are of great interest and value to the community. While other methods exist, as noted in the comparison analyses performed by the authors, the method presented here can provide improvements over existing methods at least in certain cases. The authors have also made the programs available on their website along with a guide and tutorial.

Can the authors provide any additional information about what kinds of samples might be most or least appropriate for this strategy? For example, the authors note on page 6-7 that EMfold outperforms other methods on “most of the test cases;” is there any observable pattern for which test cases are improved (or not) by this program? How does EMfold hold up if the AlphaFold protein prediction is not good, for example if large portions of the predicted sequences are intrinsically disordered? The authors note that they tested complexes with up to 10 components; is there any predicted limit to the size of complex that could be used? Some particle picking programs using neural networks do not work well with samples on grids with continuous carbon coatings; are there any such limitations for this strategy?

The authors validate their method by comparing their EMfold models to the deposited models, and comparing the EMfold models to those built by other programs. This is important and helpful information. If adopted by the field and used on new samples, we may not have deposited data to compare against. Have the authors used any of the other standard validation methods on the test set EM maps and built models such as model-map FSC, Ramachandran scores, etc?

There are some additional points that would benefit from clarification.

On page 4, I do not understand this statement: In such case, accurate segmentation of individual subunits is impossible because the low quality of the map, let alone the built model.

On page 4, I think it would be helpful to define or describe the term “main-chain probability map.” This is fully described in the Methods section, but that is much later in the manuscript.

On page 5, the authors state that their fitting strategy is faster, “without the need of long-time flexible fitting.” If the authors are suggesting significant time savings for using their strategy, it would be good to provide some benchmarks.

On page 6 I do not understand the phrase “the top-scored one among different combinations.”

On page 6, briefly define or describe TM-scores.

The term "complex model" is used ~19 times. Please define or describe what is meant by this term when first used on page 5.

The authors also frequently use the term "native structure" ("native fragment," "native template," etc). I do not understand the focus on native structure. Do we know that all 47 test cases from structural studies are in their properly folded and assembled functional form? As structural biologists, we often assume this is the case, but I'm not convinced that we know that for all deposited structures here. I don't see it as a criterion in the Methods. The authors are comparing their results to the deposited structures and, I think, referring to those as native structures. If there is a greater importance of the term "native structure," please explain it more fully.

On page 7, the authors state that they calculated the RMSD "of the aligned residues." I think this is referring to aligning the EMfold structure with the deposited structure, but it would be clearer if authors state this explicitly.

Figures

Figure 1 – In the bottom panel, left side, it looks like A2 and C2 are connected by an edge (green line). In the center of the panel, A2 and C2 do not appear to be connected. Should they be? In this panel, I would also describe the use of red and green lines in the figure legend.

Figure 2 – The coloring and labeling in this figure is not clear to me. In the top panel of 2a, I don't know why some of the ribbon diagram is in green and red. How many things are overlaid here? The main-chain probability map in transparent magenta and the PDB structure in blue are described. What are the other colors – possibly the query chain? Is there a significance in labeling the 6 boxes? This panel may need to be simplified for better visibility. The 2a bottom panel is noted in the figure legend to include the query chain colored by C alpha displacements. Is this also overlaid with the PDB structure in blue?

Figure 3a figure legend – there is an extra a in "Aaverage"

Caveat - The details of designing the programs are beyond the scope of my expertise and I do not feel qualified to assess this (sections 4.5-4.5 and 4.6).

Reviewer #2:

Remarks to the Author:

Highly accurate model building of protein complexes from intermediate resolution cryo-EM maps with deep learning-guided automatic assembling
Jiahua He, Peicong Lin, Ji Chen and Sheng-You Huang

Reviewer expertise:

Crystallography, cryo-EM, manual and automated model building, particular focus on membrane protein structure and function.

General assessment:

Model building in intermediate resolution (4-8Å) cryoEM maps remains a challenging task, even when high quality initial models are available. This will become an increasingly significant problem as intermediate resolution maps obtained by sub-tomogram averaging of cryo-ET data become more widely available. He and colleagues describe an automatic model building method, EMBuild, to address these challenging model-building scenarios. EMBuild combines several innovative approaches – first using a deep learning approach to calculate a "main chain probability map" with less noise than the

initial experimental map, which then guides rigid body and flexible fitting of templates obtained from the AlphaFold database. The results are remarkable – EMBuild is able to successfully build high quality models for the majority of the 47 selected test cases. Overall, the work outlined here is highly significant, and the results well justify the conclusions of the manuscript. The manuscript is well written, logically organized and a pleasure to read. I have just a few comments and suggestions, itemized below, for the authors to consider.

Recommendations:

- Often, intermediate resolution maps from single particle analysis are not just lower resolution overall. In my experience they more often suffer from preferred orientation and resulting anisotropy in the density map. An analysis of how robust EMBuild is to anisotropy would strengthen the analysis, as this is a challenging scenario even for manual model building at higher resolution.
- Many users of the software in the future may wish to use EMBuild on maps obtained by sub-tomogram averaging of cryoET data. It may be worth considering including some such cases in the sample set for testing.
- The measures of success in part rely on the assumption that the PDB model of the original structure is accurate. Normally I would say this is a reasonable assumption, but in the case of models built de novo into intermediate resolution maps I am not so sure. Perhaps the authors might consider analyzing a subset where high resolution crystal or cryoEM structures are also available, so that one can be confident in the accuracy of the atomic model that is being used as the point of comparison for the results of EMBuild? Alternatively, including some higher resolution test cases that have been artificially filtered back to lower resolution could be helpful in this regard.
- It would be useful to provide some analysis of computational efficiency – how long does EMBuild take to run, on what hardware, in comparison to competing methods?
- Would it be possible to use the main chain probability maps from EMBuild as input to the other tested methods? I am not sure whether this is feasible, but if so it might help to differentiate whether the extraordinary success of EMBuild is mostly due to how clean these maps are, or more due to the assembly and fitting procedure.
- Some analysis of how EMBuild deals with symmetry (and pseudosymmetry) would be helpful. Does the user have to specify the number of copies of each sequence?
- Increasingly, we are analyzing complexes purified from native sources, where we may not know with certainty which components are present in a given map. Does EMBuild have any capacity to automatically test the fit of different candidate sequences to the map?
- It is not totally clear to me what the map input for main chain probability map generation is. Is this the unsharpened map, unfiltered half maps, or something else? Does map post-processing have any effect on the quality of the resulting main chain probability map?
- In the methods it is stated that the “predicted model of each chain is cropped to have identical starting and ending residues with the sequence in the PDB structure”. While I understand why the authors have done this, for ease of comparison, it is not very realistic. In a real situation, we do not necessarily know how much of the sequence is disordered. I would suggest trimming the model in a more objective way that does not depend on prior knowledge of the solved structure. Perhaps trimming the AlphaFold template based on the pLDTT value might be one approach?
- In Figure 6, the main chain match scores are compared with C-alpha displacement, and errors are highlighted. In panel e, one region is highlighted, but looking at the model it is apparent that one of the inner helices is out of register by approximately one turn – this also qualifies in my mind as a significant model building error, but is not highlighted by the main chain match score (perhaps this is just an issue of color scale).

Reviewer #3:

Remarks to the Author:

The authors present a very interesting approach to atomic models fitted to CryoEM maps. The method

is technically sound and well explained. They show with many examples that the method is working significantly better than the state-of-the-art methods. They also make their algorithm publicly available, which is very valuable.

The manuscript is very well written and there is almost nothing to add. If anything, the authors could show the training and test learning curves in Supplementary Material.

Manuscript ID: NCOMMS-22-09612-T

Title: Highly accurate model building of protein complexes from intermediate-resolution cryo-EM maps with deep learning-guided automatic assembling

Author(s): Jiahua He; et al.

We very much appreciate the valuable comments/suggestions from the reviewers. We have conducted necessary computations/analyses and revised our manuscript accordingly. The revised parts in the manuscript are highlighted **in red**. The point-to-point responses to the comments are listed as follows.

Reviewer #1 (Remarks to the Author):

The authors have developed a program to build protein models into intermediate (4-8 Å) resolution EM maps for complexes made up of 2–10 protein components. Their approach is innovative in that it combines and builds on recent advances in model building. First, rather than modeling directly into the map, the map is used to predict a main-chain probability map using nested UNet++. Second, AlphaFold2 or other external programs are used to predict the 3D structure of the components of the protein complex of interest based on sequence. Finally, the sequence-based structural models are iteratively fit and refined in segments into the main-chain probability map. The authors trained the program on over 200 unique pairs of maps from the EMDB with corresponding models in the PDB. They then applied this approach, as well as other modeling programs, to 47 additional EMDB map entries that have corresponding PDB entries. The authors found that their approach performs as well as or better than other modeling approaches at producing results similar to the structures deposited in the PDB from the corresponding EMDB maps.

Building models into intermediate resolution EM maps is an important and significant challenge in the field. Because there is such great variation in biological samples, additional tools to address this challenge in innovative ways are of great interest and value to the community. While other methods exist, as noted in the comparison analyses performed by the authors, the method presented here can provide improvements over existing methods at least in certain cases. The authors have also made the programs available on their website along with a guide and tutorial.

Response: We thank the reviewer for reviewing our manuscript and giving the valuable comments and suggestions. We have addressed the reviewer's comments and revised our manuscript accordingly.

Can the authors provide any additional information about what kinds of samples might be most or least appropriate for this strategy? For example, the authors note on page 6-7 that EMfold outperforms other methods on “most of the test cases;” is there any observable pattern for which test cases are improved (or not) by this program? How does EMfold hold up if the

AlphaFold protein prediction is not good, for example if large portions of the predicted sequences are intrinsically disordered? The authors note that they tested complexes with up to 10 components; is there any predicted limit to the size of complex that could be used? Some particle picking programs using neural networks do not work well with samples on grids with continuous carbon coatings; are there any such limitations for this strategy?

Response: We have added a separate paragraph to provide the information about what kinds of samples might be most or least appropriate for EMBuild in the Discussion section. In brief, EMBuild is especially robust and achieves the most improvement for those cases where the proteins experience global conformational changes through relative movement of rigid domains/fragments. Although EMBuild tends to perform less satisfactorily for those cases where the AlphaFold2-predicted structure contains inaccurate regions, such limitations can be much alleviated by removing those inaccurate or intrinsically disordered parts based on their pLDDT values in the AlphaFold2-predicted structure. EMBuild can perform equally well for those samples prepared by different techniques like continuous carbon coating as long as their reconstructed 3D maps have similar resolutions. [Page 19, 2nd paragraph]

In addition, we have also added that EMBuild can be applied to any number of chains for a map, although we have tested complexes with up to 10 components for saving time on the test set of single-particle EM maps. As illustrated in the evaluation results on the newly added test set of 16 subtomogram averaging maps, EMBuild can build good models for protein complexes with up to 24 chains. [Page 22, in 1st paragraph; Pages 11-12, Section 2.4]

The authors validate their method by comparing their EMfold models to the deposited models, and comparing the EMfold models to those built by other programs. This is important and helpful information. If adopted by the field and used on new samples, we may not have deposited data to compare against. Have the authors used any of the other standard validation methods on the test set EM maps and built models such as model-map FSC, Ramachandran scores, etc?

Response: Following the reviewer's suggestion, we have added additional standard metrics including model-map FSC, Ramachandran scores, and MolProbity score to validate the built models by EMBuild and other methods on the test sets of cryo-EM and subtomogram averaging maps. The corresponding results have been added into the manuscript. [Page 10, 1st paragraph and last paragraph; Page 11, 1st paragraph; Page 14, 2nd paragraph; Page 29, 1st paragraph]

There are some additional points that would benefit from clarification.

On page 4, I do not understand this statement: In such case, accurate segmentation of individual subunits is impossible because the low quality of the map, let alone the built model.

Response: We have changed the statement to "in such case, accurate map segmentation of individual subunits is impossible because the low quality of the map, let alone reliable fitting of individual chains into the map" [Page 4, in 2nd paragraph]

On page 4, I think it would be helpful to define or describe the term “main-chain probability map.” This is fully described in the Methods section, but that is much later in the manuscript.

Response: Following the reviewer’s suggestion, we have added the description about main-chain probability map as “the main-chain probability map predicted by our deep learning model, where the density/probability value on a grid point stands for the probability of a main-chain atom existing around the grid point.” in the last paragraph of Introduction. [Page 5, in 1st paragraph]

On page 5, the authors state that their fitting strategy is faster, “without the need of long-time flexible fitting.” If the authors are suggesting significant time savings for using their strategy, it would be good to provide some benchmarks.

Response: We have removed the statement of “without the need of long-time flexible fitting.” as it may cause confusion.

On page 6 I do not understand the phrase “the top-scored one among different combinations.”

Response: We have changed the description of the phrase as “the best combination of protein chains with the highest total main-chain match score”. [Page 6, in the 1st paragraph]

On page 6, briefly define or describe TM-scores.

Response: Following the reviewer’s suggestion, we have added the description of TM-score as “...TM-score is a measure of similarity between two protein structures...”. [Page 7, in 1st paragraph]

The term “complex model” is used ~19 times. Please define or describe what is meant by this term when first used on page 5.

Response: We have added a description of the term “complex model” as “the model of the protein complex structure, i.e. complex model,” in Page 5. [Page 5, in 1st paragraph]

The authors also frequently use the term “native structure” (“native fragment,” “native template,” etc). I do not understand the focus on native structure. Do we know that all 47 test cases from structural studies are in their properly folded and assembled functional form? As structural biologists, we often assume this is the case, but I’m not convinced that we know that for all deposited structures here. I don’t see it as a criterion in the Methods. The authors are comparing their results to the deposited structures and, I think, referring to those as native structures. If there is a greater importance of the term “native structure,” please explain it more fully.

Response: We agree with the reviewer that the term “native structure” may not be suitable

here. We have now changed the term “native structure” to “deposited PDB structure” or “PDB structure”. [e.g. Page 8, last paragraph, etc.]

On page 7, the authors state that they calculated the RMSD “of the aligned residues.” I think this is referring to aligning the EMfold structure with the deposited structure, but it would be clearer if authors state this explicitly.

Response: Following the reviewer’s suggestion, we have now changed the statement to “the RMSD stands for the root mean square deviation of the aligned residues between the built complex model and the PDB structure.” [Page 7, in 1st paragraph]

Figures

Figure 1 – In the bottom panel, left side, it looks like A2 and C2 are connected by an edge (green line). In the center of the panel, A2 and C2 do not appear to be connected. Should they be? In this panel, I would also describe the use of red and green lines in the figure legend.

Response: Yes, the reviewer is correct. We have corrected the node names in Figure 1. We have also described the use of red and green lines in the figure caption as “Two poses are connected (green line) if the clash score is below a certain threshold, and are disconnected (red break line) if the clash score is above the threshold.” [Page 38, Figure 1]

Figure 2 – The coloring and labeling in this figure is not clear to me. In the top panel of 2a, I don’t know why some of the ribbon diagram is in green and red. How many things are overlaid here? The main-chain probability map in transparent magenta and the PDB structure in blue are described. What are the other colors – possibly the query chain? Is there a significance in labeling the 6 boxes? This panel may need to be simplified for better visibility. The 2a bottom panel is noted in the figure legend to include the query chain colored by C alpha displacements. Is this also overlaid with the PDB structure in blue?

Response: To avoid confusion, we have now used a consistent coloring style for all the panels in Figure 2a. That is, the main-chain probability map is in transparent magenta, the PDB structure is in blue, and the query chain is colored by C alpha displacements. In addition, we have removed the 6 boxes for better visibility. The figure caption was also improved accordingly [Page 39, Figure 2]

Figure 3a figure legend – there is an extra a in “Aaverage”

Response: It was corrected. [Page 40, Figure 3a caption]

Caveat - The details of designing the programs are beyond the scope of my expertise and I do not feel qualified to assess this (sections 4.5-4.5 and 4.6).

Response: Thanks.

Reviewer #2 (Remarks to the Author):

Highly accurate model building of protein complexes from intermediate resolution cryo-EM maps with deep learning-guided automatic assembling
Jiahua He, Peicong Lin, Ji Chen and Sheng-You Huang

Reviewer expertise:

Crystallography, cryo-EM, manual and automated model building, particular focus on membrane protein structure and function.

General assessment:

Model building in intermediate resolution (4-8Å) cryoEM maps remains a challenging task, even when high quality initial models are available. This will become an increasingly significant problem as intermediate resolution maps obtained by sub-tomogram averaging of cryo-ET data become more widely available. He and colleagues describe an automatic model building method, EMBuild, to address these challenging model-building scenarios. EMBuild combines several innovative approaches – first using a deep learning approach to calculate a “main chain probability map” with less noise than the initial experimental map, which then guides rigid body and flexible fitting of templates obtained from the AlphaFold database. The results are remarkable – EMBuild is able to successfully build high quality models for the majority of the 47 selected test cases. Overall, the work outlined here is highly significant, and the results well justify the conclusions of the manuscript. The manuscript is well written, logically organized and a pleasure to read. I have just a few comments and suggestions, itemized below, for the authors to consider.

Response: We thank the reviewer for reviewing our manuscript and giving the positive comments and valuable suggestions. We have conducted the suggested analysis and revised our manuscript accordingly.

Recommendations:

- Often, intermediate resolution maps from single particle analysis are not just lower resolution overall. In my experience they more often suffer from preferred orientation and resulting anisotropy in the density map. An analysis of how robust EMBuild is to anisotropy would strengthen the analysis, as this is a challenging scenario even for manual model building at higher resolution.

Response: Following the reviewer’s suggestion, we have added a separate section to analyze the impact of anisotropy in EM maps on the model building by EMBuild. In brief, although the map anisotropy can cause challenges to model building, EMBuild handled the anisotropy issue well by taking advantage of the predicted main-chain probability map and the AlphaFold2-predicted protein structures. [Page 15, Section 2.7; Page 45, Figure 8]

- Many users of the software in the future may wish to use EMBuild on maps obtained by sub-tomogram averaging of cryoET data. It may be worth considering including some such cases in the sample set for testing.

Response: Following the reviewer's suggestion, we have tested EMBuild on a new test set of 16 maps obtained by sub-tomogram averaging of cryoET data. Similar improvement trends of EMBuild over other methods were also observed in the cryoET test set. We have added a separate section to present the corresponding results. [Pages 11-12, Section 2.4; Page 22, 2nd paragraph; Page 43, Figure 6]

- The measures of success in part rely on the assumption that the PDB model of the original structure is accurate. Normally I would say this is a reasonable assumption, but in the case of models built de novo into intermediate resolution maps I am not so sure. Perhaps the authors might consider analyzing a subset where high resolution crystal or cryoEM structures are also available, so that one can be confident in the accuracy of the atomic model that is being used as the point of comparison for the results of EMBuild? Alternatively, including some higher resolution test cases that have been artificially filtered back to lower resolution could be helpful in this regard.

Response: We agree with the reviewer that the associated PDB structure of a map may contain errors due to its intermediate resolution. Following the reviewer's suggestion, we have built the EMBuild models from intermediate resolution maps and then evaluated the models against higher resolution structures on two representative examples. Similar accuracy was also observed on such cases. We have added a separate section to present the results accordingly. [Pages 14-15, Section 2.6]

- It would be useful to provide some analysis of computational efficiency – how long does EMBuild take to run, on what hardware, in comparison to competing methods?

Response: Following the reviewer's suggestion, we have added a separate section to discuss about the computational efficiency of EMBuild and some other methods. [Page 18-19, Section 2.12]

- Would it be possible to use the main chain probability maps from EMBuild as input to the other tested methods? I am not sure whether this is feasible, but if so it might help to differentiate whether the extraordinary success of EMBuild is mostly due to how clean these maps are, or more due to the assembly and fitting procedure.

Response: We have used the main chain probability maps from EMBuild as input to phenix.docking_in_map and gmfit. As expected, phenix.docking_in_map indeed benefited from the main chain probability map. However, gmfit did not improve the model building, which is understandable because gmfit converts the input map into gaussian mixture model as the reduced representation in the fitting procedure, which may miss detailed structural information in the main-chain probability map. We have added a separate section to address

this issue. [Page 17, Section 2.10]

- Some analysis of how EMBuild deals with symmetry (and pseudosymmetry) would be helpful. Does the user have to specify the number of copies of each sequence?

Response: Following the reviewer's suggestion, we have added a separate section to investigate the role of protein symmetry in model building by EMBuild. It was revealed that overall the performance of EMBuild was not significantly changed after applying symmetry information on most of the test cases, although EMBuild can indeed benefit from symmetry information in a few cases. Therefore, it would be okay that users may or may not use the symmetry information of a map in real applications with EMBuild. If users treat the map as a general map, they will need to provide the structure of all protein chains. If users treat the map as a symmetric one, they may just need to provide one copy of the asymmetric unit and the symmetry information. [Pages 15-16, Section 2.8; Page 46, Figure 9]

- Increasingly, we are analyzing complexes purified from native sources, where we may not know with certainty which components are present in a given map. Does EMBuild have any capacity to automatically test the fit of different candidate sequences to the map?

Response: To address the reviewer's question, we have added a separate section to evaluate the capability of EMBuild to identify the native sequence for a given cryo-EM map from a pool of candidate sequences. It was revealed that EMBuild was able to distinguish the native sequence from the decoy sequences based on the main-chain match score. The corresponding discussion was also added accordingly. [Page 18, Section 2.11]

- It is not totally clear to me what the map input for main chain probability map generation is. Is this the unsharpened map, unfiltered half maps, or something else? Does map post-processing have any effect on the quality of the resulting main chain probability map?

Response: We are sorry for not clarifying such information. Since primary maps are used for training our main-chain prediction model, sharpened maps are recommended as the input of EMBuild for the best performance of main-chain probability prediction. Sharpened maps will result in better main-chain probability maps than unsharpened half maps. We have added a paragraph to address this issue in the Discussion section. [Pages 19, last paragraph; Page 20, 1st paragraph]

- In the methods it is stated that the "predicted model of each chain is cropped to have identical starting and ending residues with the sequence in the PDB structure". While I understand why the authors have done this, for ease of comparison, it is not very realistic. In a real situation, we do not necessarily know how much of the sequence is disordered. I would suggest trimming the model in a more objective way that does not depend on prior knowledge of the solved structure. Perhaps trimming the AlphaFold template based on the pLDTT value might be one approach?

Response: Following the reviewer's suggestion, we have re-evaluated the performance of EMBuild by trimming the AlphaFold2-predicted structure based on the pLDDT value. It was revealed that the performance of EMBuild with PDB structure trimming showed no significant difference from the case of realistic pLDDT trimming. We have added a separate section to present and discuss the corresponding results. [Pages 16-17, Section 2.9]

- In Figure 6, the main chain match scores are compared with C-alpha displacement, and errors are highlighted. In panel e, one region is highlighted, but looking at the model it is apparent that one of the inner helices is out of register by approximately one turn – this also qualifies in my mind as a significant model building error, but is not highlighted by the main chain match score (perhaps this is just an issue of color scale).

Response: We thank the reviewer for pointing out this issue. Yes, it is indeed an issue of color scale. We have adjusted the color scale, which can now identify the register error in the helix as indicated by the red box in Figure 7. [Page 44, Figure 7]

Reviewer #3 (Remarks to the Author):

The authors present a very interesting approach to atomic models fitted to CryoEM maps. The method is technically sound and well explained. They show with many examples that the method is working significantly better than the state-of-the-art methods. They also make their algorithm publicly available, which is very valuable.

Response: We thank the reviewer very much for reviewing our manuscript and giving the valuable suggestion.

The manuscript is very well written and there is almost nothing to add. If anything, the authors could show the training and test learning curves in Supplementary Material.

Response: Following the reviewer's suggestion, we have added the learning curves in Supplementary Material. [Page 24, last 2 lines; Supplementary Fig. 6]

Reviewers' Comments:

Reviewer #1:

Remarks to the Author:

The authors' revisions have addressed my previous concerns. I find the paper to be much improved in ease of reading and I find the additional analysis provided in the revision to be compelling.

Reviewer #2:

Remarks to the Author:

The authors have comprehensively addressed all of my comments, and I have nothing further to add. Congratulations on a beautiful manuscript and an impressive result on a difficult problem!